# GENERALIZED ACTIVATION VIA MULTIVARIATE PROJECTION

## ABSTRACT

Activation functions are essential to introduce nonlinearity into neural networks, with the Rectified Linear Unit (ReLU) often favored for its simplicity and effectiveness. Motivated by the structural similarity between a single layer of the Feedforward Neural Network (FNN) and a single iteration of the Projected Gradient Descent (PGD) algorithm, a standard approach for solving constrained optimization problems, we consider ReLU as a projection from $\mathbb{R}$ onto the nonnegative half-line $\mathbb{R}_+$. Building on this interpretation, we extend ReLU by substituting it with a generalized projection operator onto a convex cone, such as the Second-Order Cone (SOC) projection, thereby naturally extending it to a Multivariate Projection Unit (MPU), an activation function with multiple inputs and multiple outputs. We further provide a mathematical proof establishing that FNNs activated by SOC projections outperform those utilizing ReLU in terms of expressive power. Experimental evaluations on widely-adopted architectures further corroborate MPU's effectiveness against a broader range of existing activation functions.

## 1 INTRODUCTION

Activation functions are pivotal in neural networks. They introduce nonlinearity, enable the networks to learn complex functions and, consequently, influence both the expressivity and learnability of the model (Ramachandran et al., 2017; Hendrycks & Gimpel, 2023). Notably, many common activation functions employed in deep learning, such as the Rectified Linear Unit (ReLU), the sigmoid function, and the tanh function, are Single-Input Single-Output (SISO) functions, as they map each element of the input tensor independently to a corresponding output. While this structure is proven to be empirically effective, there raises natural questions: How to extend SISO activation functions to Multi-Input Multi-Output (MIMO) ones, and is this extension advantageous?

This paper explores this question based on the relationship between a single layer of the FNN and a single iteration of the Projected Gradient Descent (PGD) algorithm, commonly used for solving optimization problems like Quadratic Programming (QP). Specifically, a single layer of the FNN activated by ReLU can replicate a single iteration of the PGD process for linearly constrained QP problems due to their shared two-step architecture: first, a linear transformation and second, a projection operation, where ReLU is viewed as the projection from $\mathbb{R}$ to the nonnegative half line $\mathbb{R}_+$. However, we theoretically prove that any shallow FNN utilizing ReLU cannot faithfully represent a single iteration of PGD for more generalized cone programming problems, where the problem is rooted in the fact that the SISO ReLU function cannot represent the MIMO cone projection in the PGD iteration.

This observation motivates us to extend the SISO ReLU function to MIMO activation functions for a potential increase in expressive capability. We focus on the projection into the convex cone operator, named as Multivariate Projection Unit (MPU), which is the source of nonlinearity in PGD iterations. The superior expressiveness of the MPU over ReLU is both proved theoretically and validated empirically through experiments on multidimensional function fitting, illustrating the necessity for extension from SISO to MIMO activation functions. Additional experiments are conducted to compare the proposed MPU with existing activation functions on popular architectures, such as convolutional neural networks and transformers. The results demonstrate that the proposed MPU outperforms ReLU, achieving testing accuracy on par with or surpassing other existing activation functions.

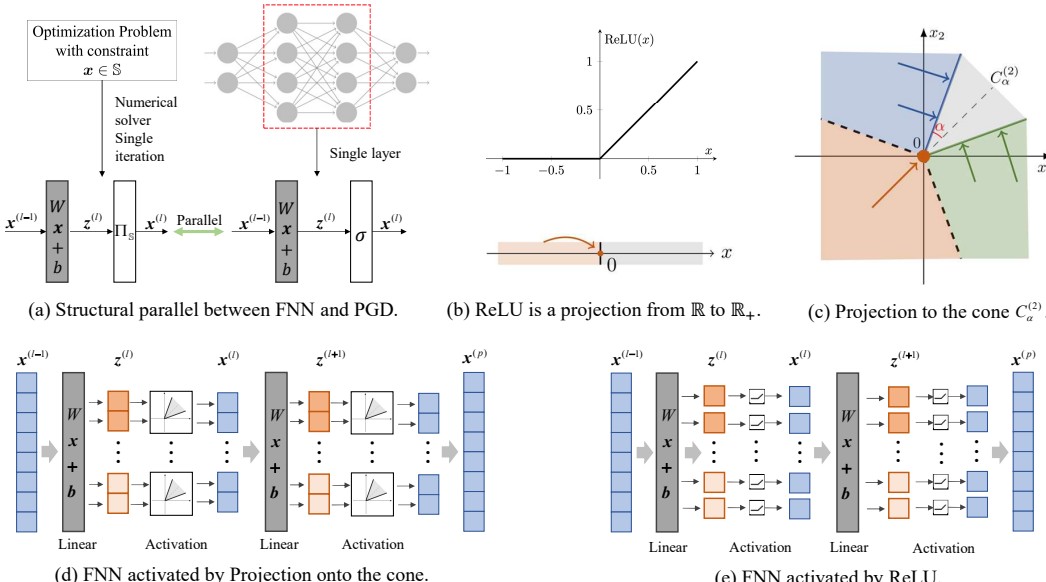

(a) Structural parallel between FNN and PGD.  (b) ReLU is a projection from $\mathbb{R}$ to $\mathbb{R}_+$.  (c) Projection to the cone $C_\alpha^{(2)}$.

(d) FNN activated by Projection onto the cone.

(e) FNN activated by ReLU.

Figure 1: The method proposed in this paper. (a) The structural similarity between a single iteration of the PGD algorithm and a single layer of the shallow FNN; (b) ReLU can be considered as the projection from $\mathbb{R}$ onto the nonnegative half line $\mathbb{R}_+$; (c) Visualization of the projection function from $\mathbb{R}^2$ onto the cone $C_\alpha^{(2)}$ in $\mathbb{R}^2$; (d) The architecture of the shallow FNN with the multivariate activation function; (e) The architecture of the shallow FNN with the ReLU activation function.

Furthermore, PGD can be viewed as a specific instance of the broader proximal gradient descent algorithm, where the projection function serves as a particular type of the proximal operator. Notably, a range of existing activation functions, including ReLU, sigmoid, tanh and softmax, which is a multivariate nonlinear function widely adopted in the transformer architecture, are already recognized as proximal operators (Combettes & Pesquet, 2019). Augmenting this body of work, we establish that the Leaky version of these proximal functions are also proximal operators leveraging the notion of Moreau's envelope. However, we note that an abundance of proximal operators remains untapped as activation functions. This observation points a future direction to the exploration of proximal operators as activation functions. A comprehensive list of both SISO and MIMO proximal operators is available at proximity-operator website (2023).

In summary, the contributions of this paper are threefold:

- Viewing ReLU as a univariate projection function, we propose to generalize the SISO ReLU function to a Multivariate Projection Unit (MPU) by substituting the projection onto $\mathbb{R}_+^n$ with the projection to more complicated shapes such as the second order cones.

- Our cone projection function possesses stronger expressive power than the ReLU activation function, which is both theoretically proved and experimentally validated.

- By drawing the connection between PGD and the FNN, it can be shown that a significant amount of the nonlinear functions adopted in literature are indeed proximal operators, which brings a future research direction on exploring new nonlinearities based on other proximal operators.

## 2 MULTIVARIATE ACTIVATION FUNCTION

In this section, we explain our rationale for interpreting the ReLU function as a projection mechanism, thereby introducing the MPU as an activation function. Moreover, we theoretically show that the proposed cone-based MPU possess greater expressive power than the single-valued ReLU function.

### 2.1 MOTIVATION: SHALLOW FNN AND PROJECTED GRADIENT DESCENT

This subsection establishes the underlying connection between one iteration of the Projected Gradient Descent (PGD) algorithm and the shallow FNN, thereby considering the ReLU function as a projection operator.

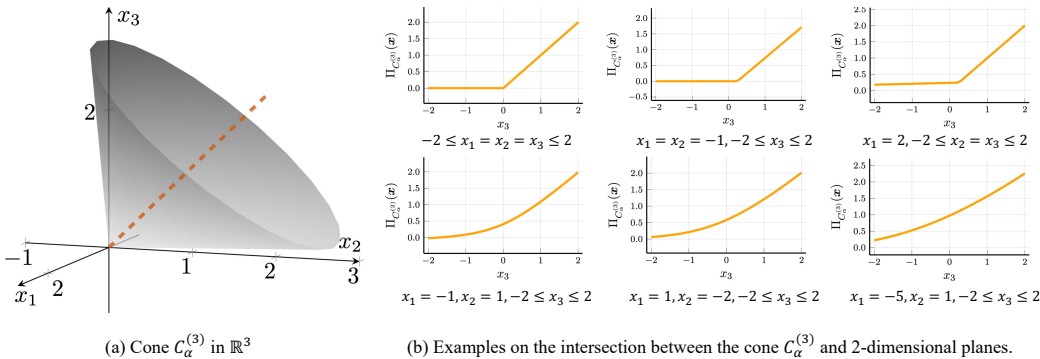

(a) Cone $C_\alpha^{(3)}$ in $\mathbb{R}^3$        (b) Examples on the intersection between the cone $C_\alpha^{(3)}$ and 2-dimensional planes.

Figure 2: The cone $C_\alpha^{(3)}$ in $\mathbb{R}^3$ and some examples on the intersection between the cone $C_\alpha^{(3)}$ and some 2-dimensional planes.

In the context of neural networks, a *neuron* represents a specific mapping defined as follows:

$$\text{Neuron}(\boldsymbol{x}) = \sigma(\boldsymbol{w}^\top \boldsymbol{x} + b),$$

where $\boldsymbol{x} \in \mathbb{R}^n$ is the input vector, $\boldsymbol{w} \in \mathbb{R}^n$ is the weight vector, $b \in \mathbb{R}$ denotes the bias term, and $\sigma : \mathbb{R} \to \mathbb{R}$ is the activation function. One of the most commonly employed activation functions in deep neural networks is the ReLU function: $\text{ReLU}(x) = \max(x, 0)$.

In modern neural networks like the classic Feedforward Neural Network (FNN), each layer comprises an array of neurons where the activation function operates independently. Formally, for the $l$-th layer of an FNN, given input $\boldsymbol{x}^{(l-1)} \in \mathbb{R}^{(n_{l-1})}$ and output $\boldsymbol{x}^{(l)} \in \mathbb{R}^{n_l}$, the transformation can be described by

$$\boldsymbol{x}^{(l)} = \text{ReLU}(W^{(l)}\boldsymbol{x}^{(l-1)} + \boldsymbol{b}^{(l)}),$$

where the ReLU function is applied element-wise. Let us define $\boldsymbol{z}^{(l)} \triangleq W^{(l)}\boldsymbol{x}^{(l-1)} + \boldsymbol{b}^{(l)}$.

On the other hand, the *projected gradient descent* method (Parikh & Boyd, 2014) is an efficacious approach for numerically solving optimization problems that take the following form:

$$\min_{\boldsymbol{x}} \frac{1}{2}\boldsymbol{x}^\top P\boldsymbol{x} + \boldsymbol{q}^\top \boldsymbol{x}, \quad \text{s.t.} \quad \boldsymbol{x} \in \mathbb{S}, \tag{1}$$

where $\boldsymbol{x}, \boldsymbol{q} \in \mathbb{R}^n, P \in \mathbb{R}^{n \times n} \succ 0$ and $\mathbb{S} \subset \mathbb{R}^n$. This canonical form can model many types of optimization problems. For example, choosing the set $\mathbb{S}$ as a polyhedron defined by a set of linear inequality constraints:

$$\boldsymbol{H}\boldsymbol{x} \leq \boldsymbol{r}, \tag{2}$$

where $\boldsymbol{H} \in \mathbb{R}^{n_{in} \times n}, \boldsymbol{r} \in \mathbb{R}^{n_{in}}$, the problem in equation 1 is the canonical form of the Quadratic Programming (QP) problem, which serves as a cornerstone optimization paradigm with extensive applicability in machine learning and control systems.

**Proposition 1** (Projected Gradient Descent (Parikh & Boyd, 2014)). *If a proper step size $\gamma > 0$ is chosen, such that $\|I - \gamma P\|_2 < 1$, then the problem in equation 1 can be solved by repeating the following two steps until convergence:*

$$\begin{aligned}
\boldsymbol{z}^{(l)} &= (I - \gamma P)\boldsymbol{x}^{(l-1)} - \gamma\boldsymbol{q}, \\
\boldsymbol{x}^{(l)} &= \Pi_{\mathbb{S}}(\boldsymbol{z}^{(l)}),
\end{aligned} \tag{3}$$

*where $\Pi_{\mathbb{S}}$ is the projection operator from $\mathbb{R}^n$ to the set $\mathbb{S}$.*

First, we have the following theorem that shows FNN with ReLU can represent the projected gradient descent for a particular form of the QP problem, and the proof is provided in Appendix A.

**Theorem 1.** *The PGD for solving the problem in equation 1 with $\mathbb{S} = \mathbb{R}_+^{n_l}$ can be represented by a single-layer FNN with ReLU activation function, i.e.,*

$$\boldsymbol{x}^{(1)} = \text{ReLU}(W^{(1)}\boldsymbol{x}^{(0)} + \boldsymbol{b}^{(1)}), \tag{4}$$

*where $W^{(1)} = I - \gamma P, \boldsymbol{b}^{(1)} = -\gamma\boldsymbol{q}$.*

**Remark 1.** *We find that some other algorithms for solving equation 1 with $\mathbb{S}$ being a polyhedral cone, such as the ADMM (Alternating Direction Method of Multiplier) algorithm (Boyd et al., 2011) and the PDHG (Primal-Dual Hybrid Gradient) algorithm (Chambolle & Pock, 2011), can also be represented by the shallow FNN with ReLU activation function.*

By this theorem, the overall PGD iteration can be exactly represented by an Recurrent Neural Network (RNN) with ReLU activation function (or equivalently, an FNN where each layer has the same weights and biases).

**Corollary 1.** *The PGD in equation 3 with $M$ iterations can be represented by an RNN with ReLU activation function, i.e.,*

$$\boldsymbol{x}^{(l)} = \mathrm{ReLU}(W\boldsymbol{x}^{(l-1)} + \boldsymbol{b}),$$

*where $\boldsymbol{x}^{(l)}$ denotes the hidden state in the l-th layer, and $W = I - \gamma P, \boldsymbol{b} = -\gamma\boldsymbol{q}$.*

Remarkably, this theorem aligns the architecture of a single-layer FNN activated by ReLU with a single iteration of PGD, shown in Fig. 1(a). This structural connection motivates us to view ReLU as a projection, thereby stimulating explorations into alternative choices for the set $\mathbb{S}$ in optimization problem equation 1.

**Example 1.** *Upon setting $\mathbb{S}$ as the polyhedron $\mathbb{P}_{-1,1} = \{\boldsymbol{x} \in \mathbb{R}^n : \boldsymbol{x}_i \in [-1, 1]\}$, the iteration in equation 3 can be represented by an FNN incorporating the hard sigmoid function (Courbariaux et al., 2015).*

Note that the particular optimization problems mentioned in Example 1 are all QP problems. In the general case where the set $\mathbb{S}$ is the polyhedron defined by the linear inequality constraints in equation 2, the optimization problem in equation 1 becomes the canonical form of the QP problem. For such instances, each iteration in PGD also contains a linear step followed by a projection step, where the projection is onto the polyhedron defined by equation 2.

Furthermore, let us define the $n$-dimensional second-order cone as $C^{(n)}$ in $\mathbb{R}^n$, i.e.,

$$C^{(n)} = \{\boldsymbol{x} \in \mathbb{R}^n \mid \|A\boldsymbol{x} + \boldsymbol{d}\|_2 \leq \boldsymbol{r}^\top\boldsymbol{x} + d\},$$

where $A \in \mathbb{R}^{n\times n}, \boldsymbol{d}, \boldsymbol{r} \in \mathbb{R}^n, l \in \mathbb{R}$. It is well-recognized that any QP problem can be recast as an equivalent Second-Order Cone Programming (SOCP) problem:

$$\min_{\boldsymbol{x}} \ \boldsymbol{c}^\top\boldsymbol{x}, \quad \text{s.t. } \boldsymbol{x} \in C^{(n)}, \tag{5}$$

but the inverse is not necessarily true. Here, the set $\mathbb{S}$ in equation 1 is chosen as $C^{(n)}$. Considering the PGD of the SOCP problem, one can observe that it still contains a linear transformation step along with a projection step. However, the projection is now to the cone $C^{(n)}$.

Each SOCP problem further admits an equivalent Semi-Definite Programming (SDP) formulation. The PGD for the SDP problem retains a similar structural composition to the FNN, except that the nonlinear transformation is now a projection onto the positive semi-definite cone $\mathbb{S}^n_+$.

In summary, let $\boldsymbol{P}_\mathbb{P}$ denote the set of optimization problems described by equation 1, with the set $\mathbb{S}$ in the constraint chosen as $\mathbb{P}$, the optimization problems mentioned above has the following inclusion relationship:

$$\underbrace{\boldsymbol{P}_{\mathbb{R}^n_+}}_{\text{Problem that can be represented by FNN+ReLU}} \subset \underbrace{\boldsymbol{P}_{H\boldsymbol{x}\leq\boldsymbol{l}}}_{\text{QP}} \subset \underbrace{\boldsymbol{P}_{C^{(n)}}}_{\text{SOCP}} \subset \underbrace{\boldsymbol{P}_{\mathbb{S}^n_+}}_{\text{SDP}}.$$

## 2.2 Method

Previously, we establish in Theorem 1 that a single-layer FNN with ReLU activation function can represent the PGD algorithm corresponding to a particular form of the QP problem. However, we find that this single-layer FNN activated by ReLU cannot represent the PGD algorithm for the SOCP problem, and the proof is reported in Section 2.3 for brevity.

Based on this observation, we propose to extend the typical univariate ReLU activation to more general projection functions, i.e., the Multivariate Projection Unit (MPU).

**Definition 1** (MPU)**.** *The (m-dimensional) Multivariate Projection Unit (MPU) is defined as the nonlinear projection function to the set $\mathbb{S} \subset \mathbb{R}^m$:*

$$\boldsymbol{P}_{\mathbb{S}} : \mathbb{R}^m \to \mathbb{S}.$$

In this paper, we discuss a special case by choosing the activation function to be the **projection onto the second-order cone** in $\mathbb{R}^m$, the projection function corresponding to the SOCP problem. Here, the dimension $m$ can be chosen to be $2, 3, \cdots$ and is not restricted to the width of each layer $n_l$. To this end, the resulting activation function becomes a multi-input multi-output function.

**Definition 2** (m-dimensional second-order cone with half-apex angle $\alpha$)**.** *We define the m-dimensional convex cone $C_\alpha^{(m)}$ in $\mathbb{R}^m$ as the convex cone centered at the origin with the axis $x_1 = x_2 = \cdots = x_m$, and half-apex angle $\alpha \in (0, \frac{\pi}{2})$, i.e.,*

$$C_\alpha^{(m)} = \{\boldsymbol{x} \in \mathbb{R}^m \mid \|\boldsymbol{h}\|_2 \le \tan(\alpha)t\},$$

*where $t = \frac{1}{\sqrt{m}}\mathbf{1}_m^\top \boldsymbol{x}, \boldsymbol{h} = \boldsymbol{x} - \frac{t}{\sqrt{m}}\mathbf{1}_m$.*

Specifically, we select $\mathbb{S}$ as $C_\alpha^{(m)}$ for $m = 2$ or $m = 3$. The cones $C_\alpha^{(2)}$ and $C_\alpha^{(3)}$ are visualized in Fig. 1(b) and Fig. 2(a) respectively. The explicit calculation of the cone $C_m^{(\alpha)}$ is derived in Appendix B. Moreover, the discussion on the different choices of the cones, as well as the analysis on the computational complexity are provided in Appendix G and Appendix H, respectively.

To incorporate the activation function $\Pi_{C_\alpha^{(m)}}(\boldsymbol{x})$ into the $l$-th layer of a neural network, the input tensor $\boldsymbol{z}^{(l)} \in \mathbb{R}^{b \times n_{l,1} \times n_{l,2} \times \cdots \times n_{l,k_l}}$ is first reshaped into a 2-dimensional vector $\tilde{\boldsymbol{z}}^{(l)} \in \mathbb{R}^{b \times \tilde{n}_{l-1}}$, where $b$ represents the batch size and $\tilde{n}_{l-1} = n_{l-1,1}n_{l-1,2}\cdots n_{l-1,k_l}$. Subsequently, the entries $\tilde{\boldsymbol{z}}^{(l)}:, p : \lfloor \frac{n_l}{m} \rfloor m + p, p = 1, \cdots, m-1$ are partitioned into $\lfloor \frac{n_l}{m} \rfloor$ sets, each processed by a projection unit $\Pi_{C_\alpha^{(m)}}$. For the residual $n_l \mod m$ dimensions, two options are available: either zero-padding $m - n_l \mod m$ dimensions followed by a projection through $C_\alpha^{(m)}$, or direct passage through the ReLU function. Empirical evidence suggests a marginal superiority of the former approach, which is thus employed in subsequent experiments. Furthermore, the explicit calculation of the projection function $\Pi_{C_\alpha^{(m)}}$ is derived in Appendix B, the implementation details are introduced in Appendix E and the choice of the cone parameters is also discussed in Appendix G.

In the following section, we theoretically discuss the expressive power of the FNN activated by the cone $C_\alpha^{(m)}$ and its relationship with that of the FNN activated by the ReLU function.

## 2.3 Expressive Capability of FNN with Cone Activation

In this subsection, we take the activation function of the FNN as the projection function to the second-order cone, and prove that the resulting neural network indeed has stronger representation power than the FNN with ReLU activation functions. The complete proof of the theorem is reported in Appendix D for brevity.

**Theorem 2** (Expressive capability for projection to cones and ReLU)**.** *The projection onto the m-dimensional cone $C_\alpha^{(m)}$ can represent the one dimensional ReLU function, i.e.,*

$$\Pi_{C_\alpha^{(m)}}(x\mathbf{1}) = \mathrm{ReLU}(x),$$

*where $\mathbf{1}$ is an m-dimensional vector of all ones.*

*On the other hand, $\forall \alpha \in (0, \frac{\pi}{2})$ and $\tan \alpha \neq \sqrt{m-1}$, no shallow FNN with ReLU activation function can faithfully represent the projection to $C_\alpha^{(m)}$. In other words, for any shallow FNN with width $d_1$ and parameters $W^{(1)} \in \mathbb{R}^{d_1 \times m}, W^{(2)} \in \mathbb{R}^{m \times d_1}, \boldsymbol{b}^{(1)} \in \mathbb{R}^{d_1}$ and $\boldsymbol{b}^{(2)} \in \mathbb{R}^2$, the following equality cannot be true for all $\boldsymbol{x} \in \mathbb{R}^m$,*

$$W^{(2)}\mathrm{ReLU}(W^{(1)}\boldsymbol{x} + \boldsymbol{b}^{(1)}) + \boldsymbol{b}^{(2)} = \Pi_{C_\alpha^{(m)}}(\boldsymbol{x}).$$

**Remark 2.** *Theorem 2 is not contradictory with the well-known Universal Approximation Theorem, which posits that any continuous function defined on a compact set can be approximated to an arbitrarily small error by a shallow FNN equipped with activation functions such as ReLU, while in*

*Theorem 2, we require the **exact** representation on the **entire space**. As a result, it is possible for a shallow FNN equipped with ReLU to approximate an MPU on a compact set to a small error. However, this may result in an explosion of the width of the FNN, which is further validated empirically in Section 4.*

**Remark 3.** *Theorem 2 further shows that a shallow FNN with a hidden dimension $d_1$ and activated by ReLU can be reparametrized to an equivalent shallow FNN, activated by projection onto $C_\alpha^{(m)}, m = 2, 3, \cdots$, with hidden dimension not exceeding $md_1$. However, for a fair comparison, we use architectures with the same width for different activation functions in the experiments later.*

## 3 EXTENSION: DESIGN ACTIVATION FUNCTIONS WITH PROXIMAL OPERATORS

The Projected Gradient Descent (PGD) algorithm presented in Section 2.1 serves as a specific instance of the more general *proximal gradient descent* algorithm (Parikh & Boyd, 2014), which is a powerful tool for numerically solving convex optimization problems, especially for those with non-smooth objective functions. Subsequently, we briefly introduce the proximal gradient descent algorithm and demonstrate that the architecture of each solver iteration aligns with that of a single layer of an FNN. Consider optimization problems taking the following form:

$$\min_{\boldsymbol{x}} \ \boldsymbol{x}^\top P \boldsymbol{x} + \boldsymbol{q}^\top \boldsymbol{x} + g(\boldsymbol{x}), \tag{6}$$

where $g(\boldsymbol{x}) : \mathbb{R}^n \to \mathbb{R}$ is a proper, lower semi-continuous convex function. To numerically solve this problem, we first introduce the notion of *proximal operator*:

**Definition 3** (Proximal operator). *Let $f : \mathbb{R}^n \to \mathbb{R}$ be a convex function. The proximal operator $\text{Prox}_f$ of $f$ is defined as a mapping from $\mathbb{R}^n$ to $\mathbb{R}^n$:*

$$\text{Prox}_f(x) = \underset{y \in \mathbb{R}^n}{\arg\min} \left\{ f(y) + \frac{1}{2} \|y - x\|^2 \right\}. \tag{7}$$

For example, the proximal operator of the indicator function:

$$\mathbb{I}_{\mathbb{S}}(\boldsymbol{x}) \triangleq \begin{cases} \boldsymbol{0}, & \boldsymbol{x} \in \mathbb{S}, \\ +\infty, & \boldsymbol{x} \notin \mathbb{S}, \end{cases} \tag{8}$$

is the projection operator to the set $\mathbb{S}$:

$$\text{Prox}_{\mathbb{I}_{\mathbb{S}}}(\boldsymbol{x}) = \Pi_{\mathbb{S}}(\boldsymbol{x}). \tag{9}$$

**Proposition 2** (Proximal gradient descent (Parikh & Boyd, 2014)). *If a proper step size $\gamma > 0$ is chosen, such that $\|I - \gamma P\|_2 < 1$, then the problem in equation 6 can be numerically solved by repeating the following two steps until convergence:*

$$\begin{aligned} \boldsymbol{z}^{(l)} &= (I - \gamma P)\boldsymbol{x}^{(l-1)} - \gamma \boldsymbol{q}, \\ \boldsymbol{x}^{(l)} &= \text{Prox}_g(\boldsymbol{z}^{(l)}). \end{aligned} \tag{10}$$

For example, all the problems introduced in Section 2.1 can be written as the following form:

$$\min_{\boldsymbol{x}} \ \boldsymbol{x}^\top P \boldsymbol{x} + \boldsymbol{q}^\top \boldsymbol{x} + \mathbb{I}_{\mathbb{S}}(\boldsymbol{x}), \tag{11}$$

and the corresponding proximal gradient descent algorithm reduces to the projected gradient descent introduced in Proposition 1. Therefore, analogous to the relationship presented in Section 2.1, the architecture of the proximal gradient descent algorithm also aligns with that of a layer of FNN, which leads to a natural question of whether certain activation functions can be interpreted as proximal operators. Indeed, this connection between activation functions and proximal operators has been observed in the literature (Combettes & Pesquet, 2019), and besides ReLU, many other activation functions are proximal operators:

**Example 2.** • *The sigmoid function is the proximal operator of the following function:*

$$\begin{cases} x \log(x) + (1 - x) \log(1 - x) - \frac{1}{2} x^2, & x \in (0, 1) \\ \qquad\qquad\qquad\qquad\qquad 0, & x \in \{0, 1\} \\ \qquad\qquad\qquad\qquad +\infty, & otherwise. \end{cases} \tag{12}$$

- *The* $\tanh$ *activation function is the proximal operator of the following function:*

$$\begin{cases} x\mathrm{arctanh}(x) + \dfrac{1}{2}(\ln(1-x^2) - x^2), & |x| < 1, \\ \\ \qquad\qquad\qquad\qquad +\infty, & otherwise. \end{cases} \tag{13}$$

- *The soft thresholding function (Pan et al., 2022) is the proximal operator of the vector 1-norm* $\|\cdot\|_1$.

- *The softmax function is the proximal operator of the negative entropy function (Combettes & Pesquet, 2019).*

More examples can be referred to Combettes & Pesquet (2019), which gives a comprehensive list of activation functions and their corresponding proximal operators.

In addition to these observations, we further provide a framework to represent the Leaky version of an activation function as a proximal operator, utilizing the concept of *Moreau envelope*.

**Definition 4** (Moreau envelope)**.** *The Moreau envelope of a proper lower semi-continuous convex function $f$ from a Hilbert space $\mathcal{V}$ to $(-\infty, +\infty]$ is defined as*

$$M_{\lambda f}(x) = \inf_{v \in \mathcal{V}} \left( f(v) + \frac{1}{2\lambda}\|v - x\|^2 \right), \tag{14}$$

*where $\lambda \in \mathbb{R}$ is a parameter of the envelope.*

Let us define the *Leaky version* of the activation function $f : \mathbb{R} \to \mathbb{R}$ as $\frac{1}{\lambda+1}f(x) + \frac{\lambda}{\lambda+1}x$, then we have the following result, where the proof is reported in Appendix C.

**Theorem 3.** *For a parameter $\lambda$, if an activation function $f : \mathbb{R} \to \mathbb{R}$ can be written as the proximal operator of a function $(\lambda + 1)g : \mathbb{R} \to (-\infty, +\infty]$, i.e., $f = \mathrm{Prox}_{(\lambda+1)g}$, then the Leaky version of $f$ is the proximal operator of the Moreau envelope $M_{\lambda g}$.*

## 4 EXPERIMENT

In this section, we perform experiments to validate the expressive power of our proposed MPU. Specifically, we employ three sets of experiments: (1) a set of experiments on multidimensional function fitting via shallow FNN to demonstrate the expressive power of our proposed MPU, (2) experiments on both CIFAR10 (Krizhevsky et al., 2009) and ImageNet-1k (Deng et al., 2009) to validate the performance of our proposed MPU on ResNet (He et al., 2016), and (3) experiments about the vision transformer Deit (Touvron et al., 2021) on ImageNet-1k (Deng et al., 2009) to demonstrate the performance of the proposed MPU. We also provide an estimation of computational complexity in terms of MACS, i.e., Multiply-Accumulate Operations. It should be noted that despite our prior theoretical analysis suggesting that a wider FNN activated by cone projection is required for accurate emulation of an FNN with ReLU activation, we retain the original architecture widths across all activation functions in our experiments for a fair comparison. The implementation details of the proposed MPU for all architectures are provided in Appendix E.

### 4.1 MULTIDIMENSIONAL FUNCTION FITTING VIA FNN

Before delving into experiments involving deep architectures, we first validate the expressive power of the proposed MPU on multidimensional function fitting, which serves as a numerical validation of our previous theoretical results. For this purpose, we compare the approximation capabilities of MPU against ReLU, PReLU, Leaky ReLU, top-50% Winner-Takes-All (WTA), MaxOut, and CReLU in shallow FNNs across two distinct functions, where the first function represents an FNN activated by a "true" multidimensional function, while the second serves as an FNN employing a "pseudo" multidimensional function for comparison:

1. A shallow FNN activated by the projection $\Pi_{C^{(2)}_{\pi/3}}(\boldsymbol{x})$ from $\mathbb{R}^2$ to the cone $C^{(2)}_{\pi/3}$ (visualized in Fig. 1(b));

2. A shallow FNN with the two activation functions, both Leaky ReLUs.

In all experiments, we use the same weight matrix, which is normalized to have a 2-norm of 1. By randomly sampling input vectors $\boldsymbol{x} \in \mathbb{R}^2$ uniformly over the square region $[-10, 10] \times [-10, 10]$, we acquire 40000 samples for training and 10000 samples for testing. The corresponding outputs are then computed by evaluating these inputs through the target functions under approximation. Each model, activated by either univariate functions or MPU, employs a shallow FNN architecture and is trained using Stochastic Gradient Descent (SGD) with a momentum of 0.9 for 50 epochs. Learning rates of $5 \times 10^{-4}$ and $0.001$ are applied for the approximation of $C_{\pi/3}^{(2)}$ and the 2-dimensional Leaky ReLU, respectively. These rates are finalized after a grid search over the set $10^{-4}, 5 \times 10^{-4}, 0.001$. Each experimental setup is executed three times under distinct random seeds $1, 2, 3$. The average loss values across these runs are then computed and illustrated in Fig. 3. Moreover, to accommodate the Single-Input Multiple-Output (SIMO) function CReLU and the Multiple-Input Single-Output (MISO) function Maxout, the structure of the corresponding FNN must be altered. In our experimentation, the input dimension for the FNN utilizing CReLU and the output dimension for the FNN employing Maxout are maintained equal to those of the other FNNs (denoted as the dimension of hidden states in Figure 3). To align with the CReLU activation function, we select the input dimension of the Maxout function to be twice that of the output dimension. As a result, the number of parameters for FNNs incorporating both CReLU and Maxout is 1.5 times that of the other FNNs.

The experimental outcomes reveal that FNNs activated by both univariate functions and MPU exhibit satisfactory performance in approximating the "pseudo" multidimensional function (the 2-dimensional Leaky ReLU). In contrast, the FNNs utilizing the SISO activation functions display a limitation in accurately approximating the projection operator $\Pi_{C_{\pi/3}^{(2)}}(\boldsymbol{x})$. This experimental result aligns with our prior theoretical results, thereby illustrating the necessity of employing MPU as activation functions.

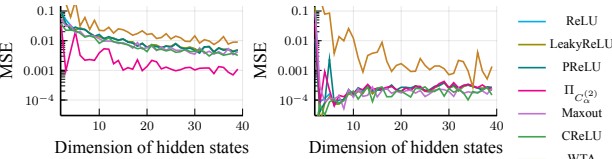

Figure 3: The Mean Squared approximation Error (MSE) of the projection $\Pi_{C_{\pi/3}^{(2)}}(\boldsymbol{x})$ and the 2-dimensional Leaky ReLU function by the FNN activated by univariate functions and the MPU plotted in a log plot w.r.t. different hidden states. *Left*: The approximation error for the FNN with the projection $\Pi_{C_{\pi/3}^{(2)}}(\boldsymbol{x})$. *Right*: The approximation error for the FNN activated by the 2-dimensional Leaky ReLU function.

## 4.2 Convolutional Neural Networks Experiments

We first testify the performance of the proposed activation function on ResNet18 (He et al., 2016) on CIFAR10 (Krizhevsky et al., 2009), and the implementation details are provided in Appendix E. The network is trained for 200 epochs with a batch size of 128, and the learning rate is initialized to 0.1 and decayed in a cosine annealing manner (Loshchilov & Hutter, 2016). The optimizer is chosen as SGD, setting weight decay to 0.0005 and the momentum to 0.9. These hyperparameters employed are directly inherited, without any modification, from the State-Of-The-Art (SOTA) ResNet training configuration (Pytorch-CIFAR-Github-repository, 2023) optimized for ReLU activation. The results are summarized in Table 1. It should be noted that the performance of the SIMO CReLU function and the MISO Maxout function is not evaluated within the classic ResNet architecture due to the requisite alternation of the network structure to integrate these functions. Given that convolutional neural networks exhibit significant structural dependencies, it remains unclear whether observed impacts are attributable to the activation functions or to the modifications in the architecture.

From the results in Table 1, we observe that our MPU, along with Leaky MPU, achieves clearly improved performance on CIFAR10, with a maximum increment of 0.18 compared to the original ReLU either in terms of the mean or the maximum test accuracy. Our performance also beats LeakyReLU and PReLU, which are also variants of ReLU, and top-50% WTA. The computational complexity of our proposed activation function is comparable to that of ReLU and Leaky ReLU. To further validate the increments of MPU, we also conduct experiments on ResNet34 and ResNet50 with ReLU. By comparing ResNet18 with MPU and ResNet34 / ResNet50 with ReLU, we observe that MPU boosts the performance of ResNet18 to a level comparable to that of ResNet34 / ResNet50

Table 1: Test accuracy of ResNet18 on CIFAR10. Test accuracy of ResNet34 and ResNet50 is also provided as a reference. Computational complexity is measured by mean MACS (over 200 epochs), i.e., Multiply–Accumulate Operations, with batch size 128. We highlight the best and the second best in bold and with underlining, respectively.

| Network | Activation | Test Accuracy (3 seeds) | Test Accuracy (Max) | Mean MACS |
|---|---|---|---|---|
| ResNet18 | ReLU | $95.35 \pm 0.10$ | $95.47$ | $71.17$G |
| | Leaky ReLU | $95.48 \pm 0.04$ (+ 0.13) | $95.53$ (+ 0.06) | $71.24$G |
| | PReLU | $94.36 \pm 0.18$ (- 0.99) | $94.60$ (- 0.75) | $71.17$G |
| | WTA | $95.36 \pm 0.02$ (+ 0.01) | $95.38$ (- 0.09) | $71.24$G |
| | **MPU ($\Pi_{C_\alpha^{(2)}}$)** | $\underline{95.51 \pm 0.10}$ (+ 0.16) | **95.60** (+ 0.13) | $71.56$G |
| | **Leaky MPU ($\Pi_{C_\alpha^{(2)}}$)** | **95.53 $\pm$ 0.03** (+ 0.18) | $\underline{95.56}$ (+ 0.09) | $71.56$G |
| | **MPU ($\Pi_{C_\alpha^{(3)}}$)** | $95.37 \pm 0.21$ (+ 0.02) | $95.53$ (+ 0.06) | $71.54$G |
| ResNet34 | ReLU | $95.62 \pm 0.02$ (+ 0.27) | $95.63$ (+ 0.16) | $148.53$G |
| ResNet50 | ReLU | $95.42 \pm 0.18$ (+ 0.07) | $95.62$ (+ 0.15) | $166.56$G |

but without largely increasing computational complexity. These results demonstrate the effectiveness of our proposed activation function on ResNet18.

We also benchmark the performance of the proposed activation function on ResNet18 (He et al., 2016) and ImageNet-1k (Deng et al., 2009). The network is trained on 8 GPU cards for 100 epochs with a batch size of 32 on each GPU, and the learning rate is initialized to 0.1 and decayed at epochs 30, 60, and 90, respectively, with a ratio of 0.1. The optimizer is chosen as SGD, setting weight decay to 0.0001 and the momentum to 0.9. These hyperparameters employed are directly inherited from the SOTA training configuration (MMPretrain-Github-repository, 2023) for the ResNet architectures utilizing ReLU activation on ImageNet, without any modifications. The results are summarized in Table 2. We observe that, as a generalization of ReLU, our MPU with $\Pi_{C_\alpha^{(2)}}$ achieves better test accuracy on ImageNet than ReLU.

Table 2: Test accuracy of ResNet18 on ImageNet.

| Activation | Test Accuracy |
|---|---|
| ReLU | $69.90$ |
| Leaky ReLU | **70.32** (+ 0.42) |
| PReLU | $68.95$ (- 0.95) |
| WTA | $66.74$ (- 3.16) |
| **MPU ($\Pi_{C_\alpha^{(2)}}$)** | $70.00$ (+ 0.10) |
| **Leaky MPU ($\Pi_{C_\alpha^{(2)}}$)** | $\underline{70.18}$ (+ 0.28) |
| **MPU ($\Pi_{C_\alpha^{(3)}}$)** | $69.64$ (- 0.26) |

Moreover, we also implement the proposed MPU on the classic vision transformer Deit-tiny (Touvron et al., 2021). The experiment result and the comparison between the proposed method and ReLU are reported in Appendix F due to space limit.

## 5 Conclusion

The paper extends the SISO activation functions in neural networks by introducing the MPU as a MIMO activation function. This extension is inspired by the structural similarity between a shallow FNN and a single iteration of the PGD algorithm. We provide rigorous theoretical proofs which show that FNNs incorporating the MPU outperform those utilizing ReLU in terms of expressive power. Moreover, by considering activation functions as proximal operators, we prove that their Leaky variants also retain this proximal property, e.g., the Leaky ReLU function. This indicates potential avenues for future research into a broader class of proximal operators, both SISO and MIMO, as activation functions.

In the experiment section, we conduct empirical validations of our proposed MPU, which include multidimensional function fitting using shallow FNNs and evaluations on CNN architectures with CIFAR10 and ImageNet-1k datasets. We observe MPU's superior performance on multidimensional function fitting and CIFAR10 image classification. However, the promotion of performance on ImageNet-1k is limited, which calls for additional investigation. Our future works include i) empirical applications of MPU on other tasks, e.g., object detection and semantic segmentation, ii) the optimizer designed for the MIMO activation functions, iii) design optimal nonlinearities in neural networks based on proximal operators.

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

## A    PROOF OF THEOREM 1

*Proof.* First, the linear transformation in equation 3 can be achieved by choosing the $W^{(1)}$ and $\boldsymbol{b}^{(1)}$ in the theorem.

For the second process, we use the following property:

$$\Pi_{\mathbb{R}_+^{n_l}}(\boldsymbol{z}^{(l)}) = \max(\boldsymbol{0}, \boldsymbol{z}^{(l)}) = \mathrm{ReLU}(\boldsymbol{z}^{(l)}),$$

where both the max operator and the ReLU function are calculated element-wise. Thus, the second process can be represented by the ReLU activation function.    □

## B    PROJECTION TO $n$-DIMENSIONAL CONE

**Theorem 4** (Projection to $n$-dimensional cone). *For any $n$-dimensional cone $C_\alpha^{(n)} \subset \mathbb{R}^n$ with center $\boldsymbol{0}_n$, half apex angle $\alpha \in [0, \frac{\pi}{2}]$ and axis passing through $(1, 1, \cdots, 1)$, the projection $\Pi_{C_\alpha^{(n)}}(\boldsymbol{x})$ can be computed via the following procedure:*

1. *Compute the height scalar $t \in \mathbb{R}$ and the vector $\boldsymbol{h} \in \mathbb{R}^n$, which is visualized in Fig. 4a:*

$$t = \frac{1}{\sqrt{n}} \mathbf{1}_n^\top x, \quad \boldsymbol{h} = \boldsymbol{x} - \frac{t}{\sqrt{n}} \mathbf{1}_n. \tag{15}$$

2. *Compute the projection: Let*

$$s = \frac{\tan(\alpha)\|\boldsymbol{h}\| + t}{\tan^2(\alpha) + 1},$$

   *then*

$$\Pi_{C_\alpha^{(n)}}(\boldsymbol{x}) = \begin{cases} \boldsymbol{x}, & \|\boldsymbol{h}\| \le \tan(\alpha)t, \\ \boldsymbol{0}, & \tan(\alpha)\|\boldsymbol{h}\| \le -t, \\ s\left(\dfrac{\mathbf{1}_n}{\sqrt{n}} + \dfrac{\tan(\alpha)\boldsymbol{h}}{\|\boldsymbol{h}\|}\right), & otherwise. \end{cases} \tag{16}$$

*Proof.* First, we compute the height scalar $t$ and the vector $h$ visualized in Fig. 4a. The expression of $t$ can be directly obtained via definition of the projection from $x$ to the axis line that passes through the point $(1, 1, \cdots, 1)$:

$$t = \frac{\mathbf{1}_n^\top \boldsymbol{x}}{\|\mathbf{1}_n\|},$$

and the vector $h$ can then be computed by subtracting the vector $t$ from $x$:

$$\boldsymbol{h} = \boldsymbol{x} - t\frac{\mathbf{1}_n}{\|\mathbf{1}_n\|}.$$

The case where $\|\boldsymbol{h}\| \le \tan(\alpha)t$ (the point $\boldsymbol{x}$ falls in the region $\mathcal{R}_1$) is trivial, since its projection is exactly itself.

The case where $\tan(\alpha)\|\boldsymbol{h}\| \le -t$ (the point $\boldsymbol{x}$ falls in the region $\mathcal{R}_3$) is also trivial, since the projection of $\boldsymbol{x}$ to the cone $C_\alpha^{(n)}$ is exactly $\boldsymbol{0}_n$.

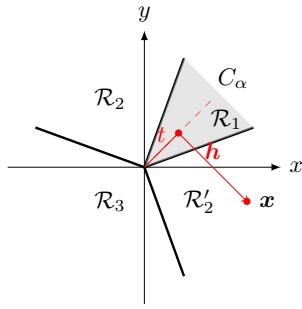

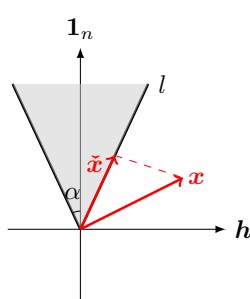

(a) The visualization concerning the meaning of the height scalar $t$ and the vector $\boldsymbol{h}$.

(b) Illustration of the plane spanned by $\mathbf{1}_n$ and $\boldsymbol{h}$.

In the following, we focus on case where $\|\boldsymbol{h}\| > \tan(\alpha)t$ and $\tan(\alpha)\|\boldsymbol{h}\| > -t$ (the point $\boldsymbol{x}$ falls into the region $\mathcal{R}_2$ and $\mathcal{R}'_2$), so that the projection of $\boldsymbol{x}$ to the cone $C_\alpha^{(n)}$ is not trivial. Let us consider the plane spanned by $\mathbf{1}_n$ and $\boldsymbol{h}$, which is visualized in Fig. 4b. The projection of $\boldsymbol{x}$ to the cone is equivalent to its projection to the boundary of the cone that lies in the plane $l$. Let us denote the projection of $x$ onto the line $l$ as the vector $\check{x}$. Then, simple geometric analysis shows that the length of the vector $\check{x}$ is

$$s \triangleq \|\check{\boldsymbol{x}}\| = \frac{\tan(\alpha)\|\boldsymbol{h}\| + t}{\tan^2(\alpha) + 1}.$$

Therefore, the coordinate of the point $\check{\boldsymbol{x}}$ in the original $n$-dimensional space is

$$\check{\boldsymbol{x}} = \|\check{\boldsymbol{x}}\| \left( \frac{\mathbf{1}_n}{\sqrt{n}} + \tan(\alpha)\frac{\boldsymbol{h}}{\|\boldsymbol{h}\|} \right) = s \left( \frac{\mathbf{1}_n}{\sqrt{n}} + \frac{\tan(\alpha)\boldsymbol{h}}{\|\boldsymbol{h}\|} \right).$$

$\square$

## C  PROOF OF THEOREM 3

*Proof.* Let $\mathcal{V} = \mathbb{R}$, and suppose $f$ is the proximal operator of $g$, then the Moreau envelope

$$M_{\lambda g}(y) = \min_{v \in \mathbb{R}} \left( g(v) + \frac{1}{2\lambda}(v - y)^2 \right),$$

and the proximal operator of $M_{\lambda g}$ is written as

$$\text{Prox}_{M_{\lambda g}}(x) = \arg\min_{y \in \mathbb{R}} \min_{v \in \mathbb{R}} \left( g(v) + \frac{1}{2\lambda}(v - y)^2 + \frac{1}{2}(y - x)^2 \right).$$

Notice that the outer minimization can be solved explicitly as

$$y^*(x) = \frac{1}{\lambda + 1}v^*(x) + \frac{\lambda}{\lambda + 1}x,$$

and the inner minimization is equivalent to

$$\min_{v \in \mathbb{R}} g(v) + \frac{1}{2(\lambda + 1)}(v - x)^2.$$

According to the definition of the proximal operator, the minimizer $v$ is the proximal operator of the function $(\lambda + 1)g$ taking value at $x$:

$$v^*(x) = \text{Prox}_{(\lambda+1)g}(x).$$

Thus, we have

$$\text{Prox}_{M_{\lambda g}}(x) = \frac{\lambda}{\lambda + 1}x + \frac{1}{\lambda + 1}\text{Prox}_{(\lambda+1)g}(x) = \frac{\lambda}{\lambda + 1}x + \frac{1}{\lambda + 1}f(x).$$

$\square$

For example, if we take $\lambda = \frac{1}{99}$, we can see that

$$\text{Prox}_{M_{1/99\mathbb{I}_{\mathbb{R}+}}}(x) = 0.99\text{ReLU}(x) + 0.01x,$$

which coincides with the Leaky ReLU.

# D    PROOF OF THEOREM 2

**Theorem 1.** *The projection to the two dimensional cone $C_\alpha$ can represent one dimensional ReLU function. However, any shallow FNN equipped with ReLU activation function cannot represent the projection to the two dimensional cone $C_\alpha$.*

*Proof.* We first prove the simple case for the cone $C_\alpha^{(2)}$ in $\mathbb{R}^2$.

First, we prove that the projection to the two dimensional cone $C_\alpha^{(2)}$ with half apex angle $\alpha \in [0, \frac{\pi}{2}]$ can represent one dimensional ReLU function. Let us consider the projection function restricted to the line $x_1 = x_2$, which is exactly the 1-dimensional ReLU, visualized in Fig. 5a.

Next, we show that any shallow FNN with arbitrary width cannot represent the 2-dimensional cone $C_\alpha^{(2)}$. Let us consider a shallow FNN with the ReLU activation function, $n_0$ inputs, $n_1$ hidden units and $n_2$ outputs and is written as:

$$\phi(x^{(0)}) \triangleq x^{(2)} = W^{(2)}\sigma(W^{(1)}x^{(0)} + b_1) + b_2, \tag{17}$$

where $\sigma = \text{ReLU}$ is calculated element-wise, $W^{(1)} \in \mathbb{R}^{n_1 \times 2}, W^{(2)} \in \mathbb{R}^{2 \times n_1}, b_1 \in \mathbb{R}^{n_1}, b_2 \in \mathbb{R}^2$. Moreover, we denote the projection function to the 2-dimensional cone $C_\alpha^{(2)}$ as $\Pi_{C_\alpha^{(2)}} : \mathbb{R}^2 \to C_\alpha^{(2)}$.

The proof is decomposed into two steps. First, we consider the case where $b_1 = 0$, and $x^{(0)}$ belongs to a compact set $\mathbb{D}_r = \{x \in \mathbb{R}^2 : \|x\|^2 \le r^2\}$ for an arbitrary $r > 0$. Notice that for the ReLU activation function,

$$\sigma(x) - \sigma(-x) = x. \tag{18}$$

Thus, we have

$$\phi(x^{(0)}) - \phi(-x^{(0)}) = W^{(2)}W^{(1)}x^{(0)}, \tag{19}$$

which is linear w.r.t. $x^{(0)}$.

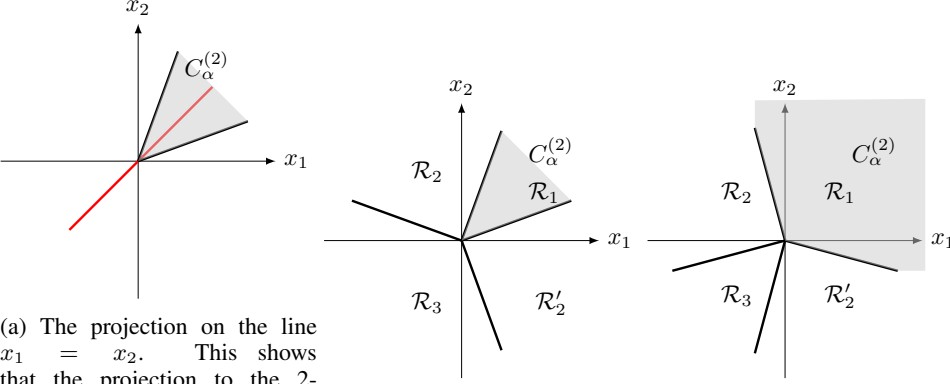

(a) The projection on the line $x_1 = x_2$. This shows that the projection to the 2-dimensional cone $C_\alpha^{(2)}$ contains the 1-dimensional ReLU.

(b) Visualization of the cone $C_\alpha^{(2)}$ where $\alpha \in [0, \frac{\pi}{4})$.

(c) Visualization of the cone $C_\alpha^{(2)}$ where $\alpha \in (\frac{\pi}{4}, \frac{\pi}{2}]$.

However, for the projection function $\Pi_{C_\alpha^{(2)}}$, we assert that it is nonlinear w.r.t. $x^{(0)}$. To see this, we shall consider the following two cases:

1. $\alpha \in [0, \pi/4)$, visualized in Fig. 5b.

   The position of the point pair $(\boldsymbol{x}, -\boldsymbol{x})$ has four different cases: the two points fall in the region pair $(\mathcal{R}_1, \mathcal{R}_3), (\mathcal{R}_2, \mathcal{R}_3), (\mathcal{R}_2, \mathcal{R}'_2)$ or $(\mathcal{R}'_2, \mathcal{R}_3)$ respectively. Due to symmetry, we

only consider the case where $x$ falls in the upper half plane. Thus,

$$
\Pi_{C_\alpha^{(2)}}(x^{(0)}) - \Pi_{C_\alpha^{(2)}}(-x^{(0)}) = \begin{cases} s\left(\dfrac{\mathbf{1}_n}{\sqrt{n}} + \tan(\alpha)\begin{bmatrix} 1 \\ -1 \end{bmatrix}\right), & x \in \mathcal{R}_2', -x \in \mathcal{R}_3, \\[2mm] x^{(0)}, & x \in \mathcal{R}_1, -x \in \mathcal{R}_3, \\[2mm] s\left(\dfrac{\mathbf{1}_n}{\sqrt{n}} + \tan(\alpha)\begin{bmatrix} -1 \\ 1 \end{bmatrix}\right), & x \in \mathcal{R}_2, -x \in \mathcal{R}_3, \\[2mm] 2s\tan(\alpha)\begin{bmatrix} -1 \\ 1 \end{bmatrix}, & x \in \mathcal{R}_2, -x \in \mathcal{R}_2', \\[2mm] -s\left(\dfrac{\mathbf{1}_n}{\sqrt{n}} + \tan(\alpha)\begin{bmatrix} 1 \\ -1 \end{bmatrix}\right), & x \in \mathcal{R}_3, -x \in \mathcal{R}_2', \end{cases}
$$
(20)

with $s = \frac{\tan(\alpha)\|h\|+t}{\tan^2(\alpha)+1}$. Notice that both $h$ and $t$ are linear w.r.t. $x^{(0)}$. Thus, the vector norm $\|h\|$ is nonlinear w.r.t. $x^{(0)}$. Therefore, we can conclude that the function $\Pi_{C_\alpha^{(2)}}(x^{(0)}) - \Pi_{C_\alpha^{(2)}}(-x^{(0)})$ is nonlinear w.r.t. the input vector $x^{(0)}$.

2. $\alpha \in (\frac{\pi}{4}, \frac{\pi}{2}]$, visualized in Fig. 5c.

The position of the point pair $(x, -x)$ has four different cases: the two points fall in the region pair $(\mathcal{R}_1, \mathcal{R}_2), (\mathcal{R}_1, \mathcal{R}_3), (\mathcal{R}_1, \mathcal{R}_2')$ or $(\mathcal{R}_2, \mathcal{R}_2')$ respectively. Due to symmetry, we only consider the case where $x$ falls in the upper half plane. Thus,

$$
\Pi_{C_\alpha^{(2)}}(x^{(0)}) - \Pi_{C_\alpha^{(2)}}(-x^{(0)}) = \begin{cases} x^{(0)} - s\left(\dfrac{\mathbf{1}_n}{\sqrt{n}} + \tan(\alpha)\begin{bmatrix} -1 \\ 1 \end{bmatrix}\right), & x \in \mathcal{R}_1, -x \in \mathcal{R}_2, \\[2mm] x^{(0)}, & x \in \mathcal{R}_1, -x \in \mathcal{R}_3, \\[2mm] x^{(0)} - s\left(\dfrac{\mathbf{1}_n}{\sqrt{n}} + \tan(\alpha)\begin{bmatrix} 1 \\ -1 \end{bmatrix}\right), & x \in \mathcal{R}_1, -x \in \mathcal{R}_2', \\[2mm] 2s\tan(\alpha)\begin{bmatrix} -1 \\ 1 \end{bmatrix}, & x \in \mathcal{R}_2, -x \in \mathcal{R}_2', \\[2mm] s\left(\dfrac{\mathbf{1}_n}{\sqrt{n}} + \tan(\alpha)\begin{bmatrix} -1 \\ 1 \end{bmatrix}\right) - x^{(0)}, & x \in \mathcal{R}_2, -x \in \mathcal{R}_1, \end{cases}
$$
(21)

with $s = \frac{\tan(\alpha)\|h\|+t}{\tan^2(\alpha)+1}$. Notice that both $h$ and $t$ are linear w.r.t. $x^{(0)}$. Thus, the vector norm $\|h\|$ is nonlinear w.r.t. $x^{(0)}$. Therefore, we can conclude that the function $\Pi_{C_\alpha^{(2)}}(x^{(0)}) - \Pi_{C_\alpha^{(2)}}(-x^{(0)})$ is nonlinear w.r.t. the input vector $x^{(0)}$.

In both cases, the resulting function $\Pi_{C_\alpha^{(2)}}(x^{(0)}) - \Pi_{C_\alpha^{(2)}}(-x^{(0)})$ is nonlinear w.r.t. the input vector $x^{(0)}$. Thus, we can see that the shallow FNN with ReLU function and bias $b_1 = 0$ cannot exactly express the projection to the 2-dimensional cone $C_\alpha^{(2)}$ in the compact set $\mathbb{D}_r$ for any arbitrary $r > 0$.

Next, we consider the case where $b_1 \neq 0$. In this case, consider the following limit:

$$
\limsup_{t \to \infty} \frac{t\sigma(W^{(1)}x^{(0)} + b^{(1)})}{t} = \sigma(W^{(1)}x^{(0)}),
$$

where the division and the limit are both taken element-wise. Therefore, equation 19 still holds if we replace $x^{(0)}$ with $tx^{(0)}$ for sufficiently large $t$.

However, one can observe from equation 20 and equation 21 that the projection to the 2-dimensional cone $C_\alpha^{(2)}$ is nonlinear w.r.t $x^{(0)}$ in both cases. Therefore, we can conclude that the shallow FNN network with ReLU activation function cannot exactly represent the projection function $\Pi_{C_\alpha^{(2)}}$ to the 2-dimensional cone $C_\alpha^{(2)}$.

Next, we prove the general case for the convex cone $C_\alpha^{(m)}$. The proof of the first part in the lemma is exactly the same as that in Theorem 2. Thus, we only focus on the second part.

First, we still consider the case where the bias of the FNN is zero. Similar to the proof of Theorem 2, it remains to find a region where the projection function $\Pi_{C_\alpha^{(m)}}$ is nonlinear w.r.t. $\boldsymbol{x}^{(0)}$. This statement is easy to verify by considering the expression of the projection function explicitly written in Theorem 4. Moreover, the extension to the case where the bias is nonzero is similar to the proof for the 2-dimensional case, and is omitted here. $\qquad\square$

# E  IMPLEMENTATION DETAILS

## E.1  EXPLICIT CALCULATION OF THE CONE

The projection to the cone $C_\alpha^{(n)}$ with $n = 2, 3, \cdots$, i.e., MPU, can be computed via the following theorem, and its proof is provided in Appendix B for brevity.

**Theorem 4** (Projection to $n$-dimensional cone). *For any $n$-dimensional cone $C_\alpha^{(n)} \subset \mathbb{R}^n$ with center $\mathbf{0}_n$, half apex angle $\alpha \in [0, \frac{\pi}{2}]$ and axis passing through $(1, 1, \cdots, 1)$, the projection $\Pi_{C_\alpha^{(n)}}(\boldsymbol{x})$ can be computed via the following procedure:*

1. *Compute the height scalar $t \in \mathbb{R}$ and the vector $\boldsymbol{h} \in \mathbb{R}^n$, which is visualized in Fig. 4a:*

$$t = \frac{1}{\sqrt{n}}\mathbf{1}_n^\top x, \quad \boldsymbol{h} = \boldsymbol{x} - \frac{t}{\sqrt{n}}\mathbf{1}_n. \tag{22}$$

2. *Compute the projection: Let*

$$s = \frac{\tan(\alpha)\|\boldsymbol{h}\| + t}{\tan^2(\alpha) + 1},$$

*then*

$$\Pi_{C_\alpha^{(n)}}(\boldsymbol{x}) = \begin{cases} \boldsymbol{x}, & \|\boldsymbol{h}\| \le \tan(\alpha)t, \\ \mathbf{0}, & \tan(\alpha)\|\boldsymbol{h}\| \le -t, \\ s\left(\dfrac{\mathbf{1}_n}{\sqrt{n}} + \dfrac{\tan(\alpha)\boldsymbol{h}}{\|\boldsymbol{h}\|}\right), & \text{otherwise.} \end{cases} \tag{23}$$

Consequently, the Leaky version of the projection to cone $C_\alpha^{(n)}$ with $n = 2, 3, \cdots$, i.e., Leaky MPU, can be computed by $0.99\Pi_{C_\alpha^{(n)}}(\boldsymbol{x}) + 0.01\boldsymbol{x}$.

The further details for the choice of the cone is discussed in Appendix G.

The multidimensional function fitting experiment is implemented in Python using the `PyTorch` package (Paszke et al., 2019). The code is self-written and is available in the submitted zip file.

The experiment of ResNet18 on CIFAR10 is mostly based on the code in `https://github.com/kuangliu/pytorch-cifar`, which reaches the highest accuracy in all the repositories for ResNet that we investigate. The hyperparameters that we use in our experiment are all directly adopted from the repository without any modifications. The code is available in the submitted zip file.

The experiment of both ResNet18 and the Deit-tiny architectures on ImageNet-1k dataset are based on the code in `https://github.com/open-mmlab/mmpretrain`. The hyperparameters that we use in our experiment are all directly adopted from the repository without any modifications.

# F  MORE EXPERIMENTS

Finally, the proposed activation function is tested on a vision transformer Deit (Touvron et al., 2021) on ImageNet-1k (Deng et al., 2009). The network is trained on 8 GPU cards for 500 epochs with a batch size of 128 on each GPU, and the learning rate is kept as 0.001 during the first 20 epochs, and then a Cosine Annealing learning rate schedule is employed in the rest of the training phase, and the

minimum learning rate is set to $10^{-5}$. We choose the optimizer as AdamW, setting weight decay to 0.05, $\epsilon$ to $10^{-8}$ and betas to $(0.9, 0.999)$. The results are summarized in Table 3.

Moreover, to further illustrate the performance of the proposed MPU on a deeper architecture, we test the ReLU, Leaky ReLU and the proposed MPU on the ResNet101 architecture on CIFAR10, and the results are summarized in Table 4.

Furthermore, as can be seen from Table 1, the inclusion of the proposed MPU slightly introduces an increase on the MACS. To facilitate a fair comparison of the proposed MPU against ReLU and other activation functions, we conducted additional experiments on the ResNet18 architecture, adjusting the number of training epochs for these functions to ensure their mean Multiply-Accumulate Operations (MACS) exceeded that of the MPU. The results are summarized in Table 5. We also slightly increase the depth of the ResNet18 architecture by adding one more basic block to form ResNet18+, increasing the resulting MACS of the ReLU, Leaky ReLU, PReLU and WTA for fair comparison. The results are shown in Table 6.

Table 3: Test accuracy of Deit-tiny on ImageNet-1k.

| Activation Functions | Top-1 Accuracy |
|---|---|
| ReLU | 74.99 |
| Proj to $C_\alpha^{(2)}$ | 74.86 |
| Proj to $C_\alpha^{(3)}$ | **75.06** |

Table 4: Test accuracy of ResNet101 on CIFAR10.

| Activation | Test Accuracy (3 seeds) | Test Accuracy (Max) |
|---|---|---|
| ReLU | $\underline{95.62 \pm 0.11}$ | 95.70 |
| Leaky ReLU | $95.57 \pm 0.34$ (- 0.05) | $\underline{95.88}$ (+ 0.18) |
| **MPU ($\Pi_{C_\alpha^{(2)}}$)** | $95.08 \pm 0.53$ (- 0.54) | 95.69 (- 0.01) |
| **Leaky MPU ($\Pi_{C_\alpha^{(2)}}$)** | $\mathbf{95.78 \pm 0.18}$ (+ 0.16) | **95.91** (+ 0.21) |
| **MPU ($\Pi_{C_\alpha^{(3)}}$)** | $95.60 \pm 0.05$ (- 0.02) | 95.64 (- 0.06) |

Table 5: Test accuracy of ResNet18 on CIFAR10. Test accuracy of ResNet34 and ResNet50 is also provided as a reference. Computational complexity is measured by mean MACS (over 200 epochs) with batch size 128. We highlight the best and the second best in bold and with underlining, respectively.

| Network | Activation | Test Accuracy (3 seeds) | Test Accuracy (Max) | Mean MACS |
|---|---|---|---|---|
| | ReLU | $95.35 \pm 0.10$ | 95.47 | 71.17G |
| | ReLU (+2 epochs) | $95.41 \pm 0.12$ (+ 0.06) | 95.51 (+ 0.04) | 71.88G |
| | Leaky ReLU | $95.48 \pm 0.04$ (+ 0.13) | 95.53 (+ 0.06) | 71.24G |
| | Leaky ReLU (+2 epochs) | $95.25 \pm 0.09$ (- 0.10) | 95.35 (- 0.12) | 71.95G |
| | PReLU | $94.36 \pm 0.18$ (- 0.99) | 94.60 (- 0.75) | 71.17G |
| ResNet18 | PReLU (+2 epochs) | $94.59 \pm 0.01$ (- 0.76) | 94.60 (- 0.75) | 71.88G |
| | WTA | $95.36 \pm 0.02$ (+ 0.01) | 95.38 (- 0.09) | 71.24G |
| | WTA (+2 epochs) | $95.23 \pm 0.11$ (- 0.12) | 95.35 (- 0.12) | 71.95G |
| | **MPU ($\Pi_{C_\alpha^{(2)}}$)** | $\underline{95.51 \pm 0.10}$ (+ 0.16) | **95.60** (+ 0.13) | 71.56G |
| | **Leaky MPU ($\Pi_{C_\alpha^{(2)}}$)** | $\mathbf{95.53 \pm 0.03}$ (+ 0.18) | $\underline{95.56}$ (+ 0.09) | 71.56G |
| | **MPU ($\Pi_{C_\alpha^{(3)}}$)** | $95.37 \pm 0.21$ (+ 0.02) | 95.53 (+ 0.06) | 71.54G |
| ResNet34 | ReLU | $95.62 \pm 0.02$ (+ 0.27) | 95.63 (+ 0.16) | 148.53G |
| ResNet50 | ReLU | $95.42 \pm 0.18$ (+ 0.07) | 95.62 (+ 0.15) | 166.56G |

Table 6: Test accuracy of ResNet18 and the modified ResNet18+ on CIFAR10. We add one more basic block in ResNet18 structure to form the ResNet18+ architecture. Test accuracy of ResNet34 and ResNet50 is also provided as a reference. Computational complexity is measured by mean MACS (over 200 epochs), with batch size 128. We highlight the best and the second best in bold and with underlining, respectively.

| Network | Activation | Test Accuracy (3 seeds) | Test Accuracy (Max) | Mean MACS |
|---|---|---|---|---|
| ResNet18 | ReLU | $95.35 \pm 0.10$ | $95.47$ | 71.17G |
| | Leaky ReLU | $95.48 \pm 0.04$ (+ 0.13) | $95.53$ (+ 0.06) | 71.24G |
| | PReLU | $94.36 \pm 0.18$ (- 0.99) | $94.60$ (- 0.75) | 71.17G |
| | WTA | $95.36 \pm 0.02$ ( + 0.01) | $95.38$ (- 0.09) | 71.24G |
| | **MPU** ($\Pi_{C_\alpha^{(2)}}$) | $\underline{95.51 \pm 0.10}$ (+ 0.16) | $\mathbf{95.60}$ (+ 0.13) | 71.56G |
| | **Leaky MPU** ($\Pi_{C_\alpha^{(2)}}$) | $\mathbf{95.53 \pm 0.03}$ (+ 0.18) | $\underline{95.56}$ (+ 0.09) | 71.56G |
| | **MPU** ($\Pi_{C_\alpha^{(3)}}$) | $95.37 \pm 0.21$ (+ 0.02) | $95.53$ (+ 0.06) | 71.54G |
| ResNet18+ | ReLU | $95.36 \pm 0.23$ (+ 0.01) | $95.63$ (+ 0.16) | 80.84G |
| | Leaky ReLU | $95.26 \pm 0.14$ (- 0.09) | $95.40$ (- 0.07) | 80.91G |
| | PReLU | $94.72 \pm 0.43$ (- 0.63) | $95.10$ (- 0.37) | 80.85G |
| | WTA | $95.25 \pm 0.04$ (- 0.10) | $95.30$ (- 0.17) | 80.90G |
| ResNet34 | ReLU | $95.62 \pm 0.02$ (+ 0.27) | $95.63$ (+ 0.16) | 148.53G |
| ResNet50 | ReLU | $95.42 \pm 0.18$ (+ 0.07) | $95.62$ (+ 0.15) | 166.56G |

# G    DISCUSSIONS ON CHOICE OF CONES

The structure of the convex cone can be determined by the following three factors:

- The vertex and axis of the cone;
- The dimension of the cone $m$;
- The half-apex angle $\alpha$.

And we shall discuss the choice of the three factors above in the following. First, the different choices of the vertex can be simply achieved by shifting the cone, which can be accomplished by the linear unit after the activation function. Thus, we fix the vertex of the cone as the origin of the $\mathbb{R}^m$ space without loss of generality. Similar idea can also be seen in most activation function design, such as the celebrated ReLU, sigmoid, tanh and so on. Moreover, we choose the axis of the cone as the line that passes through the point $(1, 1, \cdots, 1)$, for the symmetry of all input channels. The axis of the cone can also be rotated by the linear unit after the activation function.

We then discuss the impact of the dimension $m$ of the cone on the proposed MPU.

First, we show the following theoretical result on the expressive capability for different choice of $m$:

**Proposition 3.** *For $m_1 < m_2$, the single FNN layer equipped with the projection function $\Pi_{C_\alpha^{(m_1)}}$ can be be represented by that with the projection function $\Pi_{C_\alpha^{(m_2)}}$.*

The proof of the proposition is almost the same as that of Theorem 2 and is omitted here for simplicity.

Therefore, as we increase the dimension of the cone $m$, the expressive power of the single cone increases.

However, when embedding the proposed MPU in an existing architecture without changing the width of each layer, the dimension of the cone $m$ impacts the following three aspects:

- The expressive capability of each single cone (increases as $m$ increases);
- The number of cones in a single layer (decreases as $m$ increases);
- Computational complexity (increases as $m$ increases).

Therefore, there naturally raises a tradeoff among the three aspects mentioned above. And the practical choice of $m$ should be determined by the empirical performances of the experiments. For the

ResNet18 case, we perform experiments on multiple choices of $m$, and the results are summarized in Table 7. We can summarize from the results that the best choice of $m$ is $m = 2$ for ResNet18 on CIFAR10.

Table 7: Test accuracy of ResNet18 on CIFAR10 for cones with different dimensions $m$.

| Activation | Test Accuracy (3 seeds) | Test Accuracy (Max) |
|---|---|---|
| ReLU | $95.35 \pm 0.10$ | $95.47$ |
| Leaky ReLU | $95.48 \pm 0.04$ (+ 0.13) | $95.53$ (+ 0.06) |
| **2-dimensional MPU** ($\Pi_{C_\alpha^{(2)}}$) | $\underline{95.51 \pm 0.10}$ (+ 0.16) | **95.60** (+ 0.13) |
| **2-dimensional Leaky MPU** ($\Pi_{C_\alpha^{(2)}}$) | **95.53 ± 0.03** (+ 0.18) | $\underline{95.56}$ (+ 0.09) |
| **3-dimensional MPU** ($\Pi_{C_\alpha^{(3)}}$) | $95.37 \pm 0.21$ (+ 0.02) | $95.53$ (+ 0.06) |
| **4-dimensional MPU** ($\Pi_{C_\alpha^{(4)}}$) | $94.78 \pm 0.08$ (- 0.57) | $94.85$ (- 0.62) |
| **5-dimensional MPU** ($\Pi_{C_\alpha^{(5)}}$) | $94.81 \pm 0.16$ (- 0.54) | $94.92$ (- 0.55) |
| **6-dimensional MPU** ($\Pi_{C_\alpha^{(6)}}$) | $94.40 \pm 0.18$ (- 0.95) | $94.57$ (- 0.90) |

Finally, we set the half-apex angle $\alpha$ as a learnable parameter and is kept the same for each layer. Thus, by optimizing this parameter $\alpha$ via the training data, this parameter determines the compression ratio of each layer.

## H   COMPUTATIONAL COMPLEXITY

According to the explicit calculation of the MPU in Theorem 4, we summarize the worst-case computational complexity for a single $m$-dimensional MPU in Table 8. The result shows that the overall computational complexity of the general $m$-dimensional MPU grows linearly w.r.t. its dimension $m$. Moreover, the computation of the MPU can be simplified for $m = 2$, as the projection is now to a polyhedral instead of a cone. The simplified computational complexity is also shown in Table 8. Note that in practice, we replace $m$ single-input single-output activation functions with a single MPU, so that the computational complexity in Table 8 should be compared to $m$ single-input single-output activation functions. Moreover, the element-wise computation, such as plus, minus and multiplication operations can be easily parallelized on GPU.

Table 8: Worst-case computational complexity of each operations for the MPU and its relationship with the dimension $m$ of the MPU.

| Operation | Frequency | Frequency ($m = 2$) |
|---|---|---|
| $+$ | $3m - 1$ | $2m = 4$ |
| $-$ | $m$ | $m = 2$ |
| $\times$ | $m + 4$ | $4$ |
| $\div$ | $m + 3$ | $m + 3 = 5$ |
| $\sqrt{\phantom{x}}$ | $1$ | $0$ |
| Compare (ReLU) | $2$ | $3$ |
| **Overall** | $\mathbf{7m + 9}$ | $\mathbf{4m + 10 = 18}$ |

## I   RELATED WORKS

The most used activation function is ReLU (Nair & Hinton, 2010; Agarap, 2019). It largely mitigates the "gradient vanishing" problem of previously used sigmoid or tanh units (Maas et al., 2013). This issue occurs as the gradients approach 0 when the sigmoid or tanh is saturated. Recently, various activation functions have been proposed. These methods can be roughly categorized into four classes:

- *Univariate fixed activations:* LeakyReLU (Maas et al., 2013) proposes to allow for a small, non-zero gradient when the unit is not active, and Sitzmann et al. (2020) introduce periodic activation

functions to improve implicit neural representations. Dauphin et al. (2017) propose to use Gated Linear Units (GLU) and Gated Tanh Units (GTU) to improve language modeling, SiLU (Elfwing et al., 2017) propose to use sigmoid function weighted by its input, while Exponential Linear Unit (ELU) (Clevert et al., 2016) keeps the positive arguments but set constant values for negative ones, and GELU (Hendrycks & Gimpel, 2023) mitigates overfitting by introducing stochastic regularizers.

- *Univariate trainable activations:* For example, PReLU (He et al., 2015) explores to learn the parameters of the rectifiers actively, Swith (Ramachandran et al., 2017) searches for candidate activations, and Kernel-based Activation Function (KAF) (Scardapane et al., 2019) that uses an ensembled kernel function. Other variations of ReLU include PELU (Trottier et al., 2018) and FReLU (Qiu et al., 2018). Though the above and most existing activations are SISO, some multivariate activations have been proposed.

- *Multivariate fixed activations:* Concatenated ReLU (CReLU (Shang et al., 2016)) takes the concatenation of two ReLU functions as the output, which is single-input multi-output (SIMO). MaxOut (Goodfellow et al., 2013) and its variant Probabilistic MaxOut (Springenberg & Riedmiller, 2014) takes the maximum of several linear functions as the output, which is multi-input single-output (MISO). Local Winner-Take-All subnetwork (LWTA) (Srivastava et al., 2013) and top-$k$ Winner-Takes-All (WTA) (Xiao et al.) incorporate competitions to enhance multivariate activations. Other nonlinear layers such as softmax and batch norm (Ioffe & Szegedy, 2015) are also considered multivariate nonlinearities. Moreover, complex-valued activations also serve as multivariate activations (Bassey et al., 2021), e.g., multi-valued neuron (MVN) (Aizenberg et al., 1971) and cardioid activations (Virtue et al., 2017).

- *Multivariate trainable activations:* Network In Network (NIN) (Lin et al., 2014), that compute more abstract features for local patches in each convolutional layer, and its variant Convolution in Convolution (CIC) (Pang et al., 2018), which are multi-input multi-output (MIMO).

In this work, MPU is also a trainable activation that generalizes ReLU to MIMO.

