# OpenReview forum: "Generalized Activation via Multivariate Projection"
_ICLR.cc/2024/Conference — Submitted to ICLR 2024_

### Official Review · Reviewer_MyBG · 2023-10-25

**Soundness:** 3 good
**Presentation:** 3 good
**Contribution:** 2 fair
**Rating:** 8
**Confidence:** 4

**Summary:**

The authors introduce "Multivariate Projection Unit" (MPU), a novel activation function that, differently from a standard AF, takes multiple inputs and also returns multiple outputs (MIMO). The key observation is that ReLU can be seen as a a projection to the positive orthant, and linear multiplication + ReLU can be seen as a step of projected gradient descent. The MPU, instead, splits the output of the linear projection into blocks of same size (see Fig. 1), and projects each block onto a prespecified cone. This is motivated by the analogy with a PGD step on a different (more general) class of optimization problems. They validate the MPU on artificial benchmarks, a CNN on CIFAR-10 and ImageNet, and a vision transformer (only in the supplementary material), showing results that are similar or slightly better than ReLU and some ReLU variants (e.g., Leaky ReLU).

**Strengths:**

To the best of my knowledge, the MPU is novel, and it is an interesting variation of a standard ReLU with a good underlying motivation. The paper is well written, in particular the visualizations in Fig. 1 immediately show the basic idea of the paper. I am less convinced about the empirical evaluation (see below), so the practical value of the MPU is not clear.

**Weaknesses:**

I have a few general comments on the manuscript, making this (at the moment) a borderline paper for acceptance. I think most of these questions are addressable and I would be happy to increase my score.

EXPOSITION: I have found the exposition of the paper a bit strange, because there is a very long motivation for the MPU (both Section 2.1 and Section 3 serve as a motivation), but very little analysis of the MPU itself. For example: (i) there is no explicit definition of the MPU or the MPU layer; (ii) there is no discussion on how to choose the cone (they only specify which cone they use in the experiments); (iii) no discussion on how to perform the projection (which is relegated to an appendix); (iv) no discussion on the theoretical computational complexity (only an empirical computation of MACS).

RESULTS: The results do not seem very strong. Ignoring the artificial datasets, on CIFAR-10 (Table 1) it is inside 1 std of Leaky ReLU. On ImageNet (Table 2), LeakyReLU is superior but also inside 1 std. DeiT (Appendix F) is only provided for a single run and no comparison, and ReLU is still only at 0.07 distance. Also, they are not providing many important ablations and comparisons, including on the choice of the cone or its dimensionality. Many baselines are missing (see below).

RELATED WORKS: the related works section is very shallow. Strangely, they are mentioning some works on multi-input AFs which are not used in the comparisons (e.g., Maxout, CReLU). However, many other works are missing (see, e.g., https://arxiv.org/pdf/2005.00817.pdf), including winner-take-all AFs, network-in-network models. Also, complex activation functions can natively work with 2 inputs and 2 outputs and can be generalized (e.g., quaternion, octanion) for more dimensions.

**Questions:**

The major questions on the paper are related to the points above, in particular:
1. Provide a clear definition of the activation function, and add discussions on its design, including the choice of the cone and the computational complexity of the projection operation.
2. Show at least one use case where the AF provide significant improvements, either in accuracy or time.
3. Add more ablations and baselines, especially of AFs which are closely linked to the paper.

I also have a few minor additional questions:
1. In the citations there are multiple references to "MS Windows NT kernel description"?
2. P4: "the set S is a certain polyhedron", can you clarify what polyhedron or point to a specific definition in the paper.
3. I would advise to add a definition for the cone C_n.

EDIT AFTER REBUTTAL: point (1) was mostly solved. Point (2) is still standing. Point (3) was partially solved. Most minor questions were solved. I have increased my score from 6 to 8 because I believe this is an interesting research direction. However, experimental results (at the moment) are unconvincing.

---

> ### Author Response · Authors · 2023-11-17
> **Official Response to Reviewer MyBG (Part 1/2)**
>
> Thanks for your appreciation and questions! All your comments are of great help to improving our manuscript. Below are the point-to-point responses.
>
> > Q1: EXPOSITION: I have found the exposition of the paper a bit strange, because there is a very long motivation for the MPU (both Section 2.1 and Section 3 serve as a motivation), but very little analysis of the MPU itself. For example: (i) there is no explicit definition of the MPU or the MPU layer; (ii) there is no discussion on how to choose the cone (they only specify which cone they use in the experiments); (iii) no discussion on how to perform the projection (which is relegated to an appendix); (iv) no discussion on the theoretical computational complexity (only an empirical computation of MACS).
> >
> > Q4: Provide a clear definition of the activation function, and add discussions on its design, including the choice of the cone and the computational complexity of the projection operation.
>
> **A1 / A4**: We apologize for the confusion caused. Indeed, the previous structure of Section 2 is not clear enough, and thank you for pointing this out! To address these problems, we have revised the manuscript in the following aspects:
>
> 1. Add another Section 2.2 to **formally introduce** our method after the motivation Section 2.1 in the revised manuscript, where we explicitly define the MPU unit in Definition 1.
>
> 2. Add the **selection of cones** in Section G, where we mainly show the selection of three factors of the cone, including the cone's vertex and axis, the dimension of the cone, and the half-apex angle $\alpha$ of  the cone. Given all these factors, the cone is well-established.
>
> 3. **Explicit calculation** of cone projections is already derived in Theorem 4. We further refer to this theorem in Section 2.2.
>
> 4. Add the analysis for **computational complexity** analysis in Section H.
>
> We hope these revisions help to improve this paper.
>
> > Q2: RESULTS: The results do not seem very strong. Ignoring the artificial datasets, on CIFAR-10 (Table 1) it is inside 1 std of Leaky ReLU. On ImageNet (Table 2), LeakyReLU is superior but also inside 1 std. DeiT (Appendix F) is only provided for a single run and no comparison, and ReLU is still only at 0.07 distance. Also, they are not providing many important ablations and comparisons, including on the choice of the cone or its dimensionality. Many baselines are missing (see below).
> >
> > Q5: Show at least one use case where the AF provide significant improvements, either in accuracy or time.
>
> **A2 / A5**: Thank you for the helpful comments. We agree that, based on the rather limited experiments conducted, the improvements are by no means significant. However, we would like to point out that, for the sake of fairness, the same hyperparameters were maintained  across all experiments, with the sole variation being the Activation  Function (AF). Under these conditions, the enhancement in test accuracy is **typically marginal**. To substantiate our viewpoint, Table 8 of Dubey et al. [4] shows a comprehensive benchmark of the existing Single-Input Single-Output (SISO) activation functions under the same training configuration, such as ReLU, PReLU, ELU, GELU, e.t.c. From these empirical results, we can see that usually, the modification of the activation function alone cannot drastically increase the accuracy of the network, compared to the celebrated ReLU (The best performance on ResNet50 is 94.09% by CELU, ReLU achieves 93.74% already).
>
> Moreover, to further illustrate the empirical improvement of the proposed MPU, we have added more experiments on both the existing AFs, such as the Winner-Takes-All (WTA)[1], the CReLU[2] and MaxOut[3], and the proposed method on a deeper ResNet101. Both results further illustrate the superiority of the proposed MPU in most settings. Notably, the new empirical results on ResNet101 show a significant increase in the accuracy of the proposed 2-dimensional Leaky MPU, whose average accuracy reaches the highest among all. Moreover, its maximum test accuracy also achieves the highest.
>
> As for the experiments on the choice of the cone and its dimensionality, we are now performing additional experiments for further investigation. We sincerely thank you for this suggestion. However, we are still running the code, and we will report our results later before the discussion stage ends.

---

> ### Author Response · Authors · 2023-11-17
> **Official Response to Reviewer MyBG (Part 2/2)**
>
> > Q3: RELATED WORKS: the related works section is very shallow. Strangely, they are mentioning some works on multi-input AFs which are not used in the comparisons (e.g., Maxout, CReLU). However, many other works are missing (see, e.g., https://arxiv.org/pdf/2005.00817.pdf), including winner-take-all AFs, network-in-network models. Also, complex activation functions can natively work with 2 inputs and 2 outputs and can be generalized (e.g., quaternion, octanion) for more dimensions.
> >
> > Q6: Add more ablations and baselines, especially of AFs which are closely linked to the paper.
>
> **A3 / A6**: We sincerely thank the reviewer for the suggestions. We have modified the related works section to include more representing approaches and divided them into four categories for clarity. Moreover, we have added more comparison experiments to Section 4, including winner-takes-all [1], Maxout [2], and CReLU [3]. We can observe that our MPU still beats these methods in the tested settings.
>
> Moreover, we assume that by "complex activation functions", you are referring to the activation functions of the complex-valued neural networks. We have reviewed some related papers briefly added the related contents to the literature review section.
>
> > Q7: In the citations there are multiple references to "MS Windows NT kernel description"?
>
> **A7**: We apologize for the confusion, and we have corrected the citations in the revised version.
>
> > Q8: P4: "the set S is a certain polyhedron", can you clarify what polyhedron or point to a specific definition in the paper.
>
> **A8**: We apologize for the confusion. After thorough thinking, we found that the corresponding example is too complicated to explain and is not rigorous enough, so we removed this example for the simplicity of the presentation.
>
> > Q9: I would advise to add a definition for the cone C_n.
>
> **A9**: Thank you for the comment, and we apologize for the confusion. In the revised manuscript, we have added a formal definition in Section 2.1 to enhance the clarity of the paper.
>
> Thanks again for your time, and we would be happy to answer any additional queries you may have!
> # References
>
> [1] Xiao, C., Zhong, P., & Zheng, C. Enhancing adversarial defense by k-winners-take-all.
>
> [2] Goodfellow, I. J., Warde-Farley, D., Mirza, M., Courville, A., & Bengio, Y. Maxout Networks.
>
> [3] Shang, W., Sohn, K., Almeida, D., & Lee, H. Understanding and Improving Convolutional Neural Networks via Concatenated Rectified Linear Units.
>
> [4] Dubey, S. R., Singh, S. K., & Chaudhuri, B. B. Activation Functions in Deep Learning: A Comprehensive Survey and Benchmark.

---

> ### Comment · Reviewer_MyBG · 2023-11-20
>
> I thank the authors for the answers. I think the exposition in the paper is now clearer, apart from some important details which can be understood from the paper but are mentioned explicitly only in the supplementary material (e.g., splitting the vector into chunks, etc.). I have also appreciated the addition of some baselines. I agree with the authors that only changing the AF may not result in significant improvements, however, this also makes the paper weaker because it's not clear what is, then, the overall benefit. Still, I have raised my vote to 8 mostly because I think this is an interesting research direction and the paper is overall well written and a good methodological contribution.

---

> > ### Author Response · Authors · 2023-11-21
> >
> > Thank you so much for your constructive feedback and the recognition of the potential within our research direction. Your comments have provided us with valuable insights! We are committed to further investigating and expanding the capabilities of the proposed MPU and are motivated to continue exploring this direction.
> >
> > Moreover, we have explained several important details (e.g., splitting the vector into chunks) in the main text for a better understanding. Thanks again for your suggestions!

---

> ### Author Response · Authors · 2023-11-21
>
> We want to supplement the answer **A2 / A5** a little.
>
> First, we want to explain the reason why the performance of the related NIN, CReLU, and Maxout functions are not evaluated on the classic ResNet architecture. To integrate these functions to CNNs like ResNet, we need to alternate the network architectures. Given that convolutional neural networks exhibit significant structural dependencies, it remains unclear whether observed impacts are attributable to the activation functions or to the modifications in the architecture. However, we have still tested the performance of the CReLU function and the Maxout function in Section 4.1 by modifying the structure of the shallow FNN. Though the FNNs with both functions use 1.5 times the number of parameters than the rest of the FNNs, their performance is still similar to those with SISO functions. This result further demonstrates the superiority of the proposed MPU.
>
> Moreover, we have accomplished the experiments on the higher dimensional cone, i.e., $m=4, 5, 6$, and the result is reported in Table 7 of the revised manuscript. The result shows the superiority of the 2-dimensional MPU over those in higher dimensions. We hope the result further addresses your concerns!

---

### Official Review · Reviewer_WKdp · 2023-10-29

**Soundness:** 3 good
**Presentation:** 2 fair
**Contribution:** 2 fair
**Rating:** 6
**Confidence:** 4

**Summary:**

The paper expands the SISO activation functions in neural networks by introducing the MPU as a MIMO activation function. This expansion is motivated by the structural resemblance between a shallow FNN and a single iteration of the PGD algorithm. Experiments test the effectiveness of the proposed approach.

**Strengths:**

1.	The idea that choose the activation function to be the projection onto the convex cone is very interesting.
2.	The paper provides some theoretical proofs.
3.	Some experiments demonstrate the effectiveness of the proposed MIMO activation function.

**Weaknesses:**

1.	The organization and writing of the paper need to be improved.
2.	The proposed Theorem 1 is not so rigorous.
3.	The experiments are not enough.

**Questions:**

1.	The projected gradient descent (PGD) algorithm in proposition 1 has many iterative steps. However, the Theorem1 has a single layer, which corresponds a single iteration of the PGD algorithm. The structural connection is weak.
2.	In Theorem 1, it seems that it needs to remove W^{(2)}=I, b^{(2)}=0. We only need  W^{(1)}, b^{(1)}, and ReLU activation function to represent a single step of PGD algorithm.
3.	How to set \alpha in Theorem 1 and Proposition 2 in the network training? It is especially to the convolutional network. It is better to give some discussion on the setting of \alpha.
4.	The paper only gives the derivations with fully-connected feedforward networks. It does not present the skip-connection. Since the experiments use ResNets, it is better to discuss this issue.
5.	The paper only tests the proposed MPU with ResNet18. It is not a very deep network.  Is the proposed MPU unable to train deep networks? It is best to give an experiment to train deep networks, such as resnet101? This is very critical, which determines whether the proposed MPU can be applied in practice.
6.	The organization of section 2 needs to be improved. It is hard to follow.
7.	When referencing equations, it is better to use “\eqref”.

---

> ### Author Response · Authors · 2023-11-17
> **Official Response to Reviewer WKdp (Part 1/2)**
>
> Thanks for your appreciation of this work and the careful reading! All your advice is of great help to us. Below are the point-to-point responses.
>
> > Q1: The organization and writing of the paper need to be improved.
>
> **A1**: We apologize for not organizing the paper properly, which led to some confusion. We have made some modifications to the previous manuscript. Some significant changes in relation to the organization and clarity of the paper are listed below:
>
> 1. We have added Section 2.2 to **formally introduce** the proposed method. We believe that this division allows the reader to quickly locate the method that we propose and better follow the discussion.
>
> 2. We have added the **selection of cones** in Section G, where we mainly show the selection of three factors of the cone, including the cone's vertex and axis, the dimension of the cone, and the half-apex angle $\alpha$ of  the cone. Given all these factors, the cone is well-established.
>
> 3. To increase the readability of the paper, we have restructured Appendix Sections E and G to illustrate the three aspects of the MPU's implementation clearly.
>
> 4. We have introduced the **rigorous definition** of the $n$-dimensional cones, the MPU, and the projection cones $C_\alpha^{(m)}$.
>
> > Q2: The proposed Theorem 1 is not so rigorous.
>
> **A2**: Thank you for the careful reading and the valuable comment. We apologize for not writing the theorem rigorously. Is the confusion due to comparing a single PGD iteration with a shallow FNN featuring a linear-activation-linear structure? This comparison may be misleading, as the two do not necessarily correlate directly. We have refined Theorem 1 and all the other related illustrations to connect a single PGD iteration with a single FNN layer. Furthermore, we have also explicitly written the expression of the FNN for better clarity.
>
> We are not sure if this resolves your concern. If not, we would be happy to answer more questions!
>
> > Q3: The experiments are not enough.
>
> **A3**: Thank you. We have performed more experiments suggested by the reviewers, such as benchmarking more existing activation functions, including Winner-takes-all [1], Maxout [2], and CReLU [3], and investigating the performance of the proposed MPU on deeper network architectures, such as ResNet101. The results are reported in the revised manuscript in Sections 4 and F.
>
> Moreover, we have also performed additional experiments on the MPU with higher dimensional cones, i.e., $m=4, 5, 6$, to show the reason why we chose small $m$, and the result is reported in Table 7. The result shows the superiority of the 2-dimensional MPU over those in higher dimensions. We sincerely thank you for this suggestion.
>
> > Q4: The projected gradient descent (PGD) algorithm in proposition 1 has many iterative steps. However, the Theorem1 has a single layer, which corresponds a single iteration of the PGD algorithm. The structural connection is weak.
>
> **A4**: Thank you for your valuable comment. First, we want to stress that this structural connection only serves as the **motivation** for our derived activation function. Moreover, we apologize for not making it clear in the paper, and we want to briefly explain the connection here. First, the connection between a single layer of the PGD and a single FNN layer is established in Theorem 1. Moreover, for the neural network with multiple layers, an RNN (or, equivalently, an FNN where the weights and biases of each layer are equivalent) exactly corresponds to the PGD iteration. To further clarify the relationship for multiple iterations, we have also added Corollary 1 in the revised manuscript.
>
> Moreover, this connection has also been noticed by some other literature, especially in the field of Learning to Optimize (L2O); see [4] for a comprehensive review. For example, Gregor and Lecun [5] leverage this connection and construct a neural network to implement a truncated form of the Iterative Shrinkage and thresholding Algorithm (ISTA), a special case of the proximal gradient descent algorithm introduced in Proposition 2, where the function $g$in equation 6 is chosen as the vector 1-norm $\|\cdot\|_1$.
>
> > Q5: In Theorem 1, it seems that it needs to remove $W^{(2)}=I, b^{(2)}=0$. We only need $W^{(1)}, b^{(1)}$, and ReLU activation function to represent a single step of PGD algorithm.
>
> **A5**: Thanks for your comment, and we apologize for the confusion caused. Indeed, it is more proper to compare a single iteration of the PGD with a single FNN layer. We have modified Theorem 1 and explicitly derived the expression of the FNN layer to better illustrate this point.

---

> ### Author Response · Authors · 2023-11-17
> **Official Response to Reviewer WKdp (Part 2/2)**
>
> > Q6: How to set $\alpha$ in Theorem 1 and Proposition 2 in the network training? It is especially to the convolutional network. It is better to give some discussion on the setting of $\alpha$ .
>
> **A6**: Actually, the $\alpha$ in Theorem 1 and Theorem 2 do not have the same meaning, and we are deeply sorry for the confusion caused. The $\alpha$ in Theorem 1 means the step size of the PGD process, and the $\alpha$ in Theorem 2 denotes the half-apex angle of the second-order cone. In the revised manuscript, we have changed the $\alpha$ in Theorem 1 and Proposition 2 to $\gamma$ for clarity.
>
> For the parameter $\gamma$ in Theorem 1 and Proposition 2, we solely leverage this expression to discuss the correspondence between a single layer of FNN and a single iteration of PGD and show that these two terms are exactly the same if the weights and biases of the FNN are chosen as the expression in Theorem 1. However, this $\gamma$ is not considered in the training process. We still directly learn the weights and biases of the FNN during training.
>
> > Q7: The paper only gives the derivations with fully-connected feedforward networks. It does not present the skip-connection. Since the experiments use ResNets, it is better to discuss this issue.
>
> **A7**: The reason why we consider FNN in Section 2 is merely for theoretical analysis. To the best of our knowledge, most existing results on the expressive capability of neural networks also focus on shallow FNN or CNN, such as multiple versions of the universal approximation theorem summarized in Wikipedia: universal approximation theorem or the univariate approximation theorem for the CNN [6]. However, the expressive capability of ResNet architecture is less studied in theory. In our paper, we analyze the expressive capability for a shallow FNN theoretically. As for ResNet with skip-connections, we empirically illustrate in the experiment section that the incorporation of MPU  also pushes the ResNet toward enhanced performance.
>
> > Q8: The paper only tests the proposed MPU with ResNet18. It is not a very deep network. Is the proposed MPU unable to train deep networks? It is best to give an experiment to train deep networks, such as resnet101? This is very critical, which determines whether the proposed MPU can be applied in practice.
>
> **A8**: We agree that the performance of the proposed method on a deeper network is also critical. To show this, we have performed benchmarking of the proposed method on ResNet101 architecture, and the result is shown in Section F. We can observe that our MPU still introduces higher accuracy when compared to baseline methods.
>
> > Q9: The organization of section 2 needs to be improved. It is hard to follow.
>
> **A9**: Thank you for your comment, and we apologize for any confusion. To enhance the clarity of Section 2, we have added Section 2.2 after the motivation section to **formally describe** our MPU and the definition of the $n$-dimensional cone. Other than that, we have revised the structure of Section E to clearly show the three aspects of the MPU's implementation and added Section G to discuss the selection of cones. We hope these modifications enhance the readability of the overall paper.
>
> > Q10: When referencing equations, it is better to use “\eqref”.
>
> **A10**: Thanks for your kind suggestion. However, all of the references of equations are already "\eqref". We assume that you are referring to using the format "(1)", where 1 is the equation number, to refer to a certain equation. This is indeed the original reference format provided by the amsmath package of LaTeX. However, in the ICLR template, this command is redefined as "equation 1" or "Equation 1", which is the kind of format in the paper.
>
> Thanks again for your valuable comments! Please let us know if you have any more questions.
>
> # References
> [1] Xiao, C., Zhong, P., & Zheng, C. Enhancing adversarial defense by k-winners-take-all.
>
> [2] Goodfellow, I. J., Warde-Farley, D., Mirza, M., Courville, A., & Bengio, Y. Maxout Networks.
>
> [3] Shang, W., Sohn, K., Almeida, D., & Lee, H. Understanding and Improving Convolutional Neural Networks via Concatenated Rectified Linear Units.
>
> [4] Chen, T., Chen, X., Chen, W., Wang, Z., Heaton, H., Liu, J., & Yin, W. Learning to Optimize: A Primer and A Benchmark.Journal of Machine Learning Research.
>
> [5] Gregor, K., & Lecun, Y. Learning Fast Approximations of Sparse Coding.
>
> [6] Yarotsky, D. Universal approximations of invariant maps by neural networks.

---

> > ### Comment · Reviewer_WKdp · 2023-11-21
> > **Thank you for answers.**
> >
> > Thank you for your detailed answers. Most of my concerns have been addressed. So I raise my score. It seems that the advantage of MPU in deep network is not obvious.

---

> > > ### Author Response · Authors · 2023-11-22
> > >
> > > Thank you so much for your appreciation of this work! This means a lot to us.

---

### Official Review · Reviewer_L1wf · 2023-10-30

**Soundness:** 3 good
**Presentation:** 4 excellent
**Contribution:** 3 good
**Rating:** 6
**Confidence:** 3

**Summary:**

Most activations functions employed in DNNs are Single Input Single Output (SISO). One can naturally ask whether we can gain by using a Multiple Input Multiple Output (MIMO) activation function instead. Of course, the answer depends on how we design such MIMO activations. This paper explores a specific strategy for designing MIMO activations: first based on projections, and more generally based on proximal functions.

The motivation the authors give for this two types of MIMO activations is that ReLU (one of the most widely used SISO activations) is a projection, and that a shallow network can be parameterized so to mimic a single iteration of Projected Gradient Descent. This is the high level observation that motivates their definition. To further motivate it mathematically, the authors prove a couple of theorems showing that ReLU can be, in a sense, mimicked by their activation, while the other direction does not.  The authors further explore proximal MIMO functions, by using proximal operator, and connecting to proximal gradient descent. They motivate it as a MIMO extension of leaky activation functions.

Finally, the authors show experiments that validate their theory and the fact that somewhat better empirical results can be obtained with their methods.

**Strengths:**

Overall, I liked the paper. It was very fun reading. In particular, the following are the main strengths:

- Very well written paper.
- Clear motivation in what they want to achieve (MIMO activation) and in how they achieve it (based on projections/proximal functions).
- Supporting theory.
- Empirical results suggesting the benefits of their approach.

**Weaknesses:**

However, there are some very substantial weaknesses:

- *[Removed due to answers by authors:]* Theory is basic and simple. But more importantly: it does not really, at the essence, establish why they function should give better results. Sure, you can set the weights just correctly so that a layer is like an iteration. But, so what? The weights are actually learned, and our goal is not to really learn an iteration...

- *[Removed due to answers by authors:]* I am a bit skeptical about the motivation. The way I see it, the whole idea in the activation is to introduce *some* nonlinearity. It does not have to be a lot, to really get a low of expressive power! Just by doing many layers, we can take a really small amount of non-linearity and make it a lot. So, not sure that MIMO really will help...

- *[Added due to answers by authors:]* Limited potential improvements: the authors observed that only using very small input-output size (namely, m=2, which replaces one input one output with two input two output) is the best you can do. Going to m=3 and above does not help. This suggest that there is very little to gain from MIMO non-linearity.

- Empirical improvement is very small, and even then not always achieved.

**Questions:**

- How are MIMO activations implemented? *[Following rebuttal:]* Still unclear on how exactly the projection itself is implemented.
- Why use small m in the experiments? *[Following rebuttal:]* OK, understood. However, this raises the question on why MPU is a good idea if only very small m is used. See new weakness above.
- Table 1: improvements in training error should be compared with cost (MACS here). With MPU you get slighly better errors with slightly more MACS. What happens with ReLU if you increase the MACS a bit, e.g. by doing a few more epochs?
Sure, you test ResNet 34 and 50  in which MACS increase. But there is more than one way to increase the MACS...
*[Following rebuttal:]* OK, doing more epochs might make sense less, but there are other ways. e.g. slightly wider network?
-*[New following rebuttal:]*  The authors use the same hyperparameters across all experiments, with the sole variation being the Activation Function (AF)." Why? And, more importantly, how the results will look if all algorithms use their best hyperparameters?

---

> ### Author Response · Authors · 2023-11-17
> **Official Response to Reviewer L1wf (Part 1/2)**
>
> Thank you for your careful reading and valuable suggestions. They are of great help in improving our manuscript. The following are the point-to-point responses.
>
> > Q1: Theory is basic and simple. But more importantly: it does not really, at the essence, establish why they function should give better results. Sure, you can set the weights just correctly so that a layer is like an iteration. But, so what? The weights are actually learned, and our goal is not to really learn an iteration...
>
> **A1**: Thank you for your insightful comment, and we are sorry for any confusion caused by the presentation of the previous manuscript. First, we would like to clarify that we **DO NOT intend to** learn an iteration of the Projected Gradient Descent (PGD) algorithm. The connection between a shallow FNN and one iteration of the projected gradient descent algorithm only serves to **motivate the usage of MPU to replace ReLU**, as in projected gradient descent, the projection can be a general projection to a convex set instead of projection to non-negative numbers (ReLU). The actual neural networks with MPU are by no means relevant to iterations of PGD.
>
> On the other hand, the potential benefit of using an MPU instead of a simple ReLU is validated both theoretically and empirically. To be specific, Theorem 2 theoretically demonstrates that any FNN with ReLU activation can be faithfully reparameterized by an FNN with MPU activation, while the vice versa is impossible regardless of the width of the ReLU network. We believe that this result theoretically guarantees that our proposed structure has the ability to give a better performance than the ReLU function. Moreover, empirical results in Section 4 also show that in the training procedure, the proposed method also gives a better result than most of the existing activation functions tested.
>
> > Q2: I am a bit skeptical about the motivation. The way I see it, the whole idea of the activation is to introduce some nonlinearity. It does not have to be a lot, to really get a low of expressive power! Just by doing many layers, we can take a really small amount of non-linearity and make it a lot. So, not sure that MIMO really will help...
>
> **A2**: Thank you so much for the valuable comments. We agree that we can increase the number of layers to enhance the expressive power of the neural network. However, we would like to first clarify that our approach, which potentially makes one layer more expressive, is **orthogonal** to the "go deeper" approach. The direction of increasing the expressive power of activation units alone is also explored in various activation function research, such as PReLU [1], CReLU [2], and so on.
>
> In numerical experiments, we find that the best test accuracy (Table 1) achieved by a ResNet18 with MPU is similar to that of a ResNet34 using ReLU, which could be an indicator to show in certain settings, it is much more economical to use a "shallower" network with more "non-linear" activation functions, rather than using a "deeper" network with small "non-linear" activations, such as ReLU.
>
> Moreover, we also added another experiment on the deeper architecture ResNet101, whose result is reported in Appendix Section G. The result also shows that by **combining the benefit** of a deep network and an activation function with more nonlinearity (the proposed Leaky MPU), the test accuracy achieves the highest among all the architectures tested.
>
> > Q3: Empirical improvement is very small, and even then not always achieved.
>
> **A3**: Thank you for your question. We agree that, based on the rather limited experiments conducted, the improvements are by no means significant. However, we would like to point out that, for the sake of fairness, the same hyperparameters were maintained  across all experiments, with the sole variation being the Activation  Function (AF). Under these conditions, the enhancement in test accuracy is **typically marginal**. To substantiate our viewpoint, Table 8 of Dubey et al. [3] shows a comprehensive benchmark of the existing Single-Input Single-Output (SISO) activation functions under the same training configuration, such as ReLU, PReLU, ELU, GELU, e.t.c. From these empirical results, we can see that usually, the modification of the activation function alone cannot drastically increase the accuracy of the network, compared to the celebrated ReLU (The best performance on ResNet50 is 94.09% by CELU, ReLU achieves 93.74% already).
>
> Moreover, to further illustrate the empirical improvement of the proposed MPU, we have added more experiments on both the existing AFs, such as the Winner-Takes-All (WTA)[4], the CReLU[2] and MaxOut[5], and the proposed method on a deeper ResNet101. Both results further illustrate the superiority of the proposed MPU in most settings.

---

> ### Author Response · Authors · 2023-11-17
> **Official Response to Reviewer L1wf (Part 2/2)**
>
> > Q4: How are MIMO activations implemented?
>
> **A4**: Thank you for the valuable question. For the implementation of the MIMO activation functions, especially the proposed MPU, three aspects are included in the paper, i.e.,
>
> 1.  The **selection of cones** is demonstrated in Section G. We mainly show the selection of three factors of the cone, including the cone's vertex and axis, the dimension of the cone, and the half-apex angle $\alpha$ of  the cone. Given all these factors, the cone is well-established.
>
> 2.  The **integration of MPU** into neural networks in Section E. Specifically, for the $m$-dimensional MPU, we flatten the input and divide them into several length-$m$ blocks. After passing each block through MPU, they are concatenated and recovered to the original shape.
>
> 3. Explicit **calculation of cone projections** in Theorem 4.
>
> To increase the readability of the paper, we have restructured Appendix Sections E and G to illustrate these three aspects clearly.
>
> > Q5: Why use small m in the experiments?
>
> **A5**: Thanks for the question. The dimension $m$ of the MPU serves as a tradeoff between the expressive power of a single MPU (a stronger expressive capability for a larger $m$) and the total number of MPUs. This is because the inputs are flattened and divided into several length-$m$ blocks, so the number of MPU decreases as $m$ increases. After comparing the empirical results in our experiments, we found that the modification from ReLU to the 2-dimensional cone benefits the performance the most, and increasing m to 3 or larger may not result in a better performance.
>
> Indeed, it is beneficial to discuss the choice of cone (including its dimension m), and thank you for pointing this out. Therefore, we have added Section G to discuss the choice of cone and conducted a series of experiments in Section 4, which shows the superiority of 2-dimensional MPUs to 3-dimensional ones with ResNet18. Moreover, we have added the experimental results for $m=4, 5, 6$, which are reported in Table 7. The result also shows the superiority of the MPUs with small dimensions $m$.
>
> > Q6: Table 1: improvements in training error should be compared with cost (MACS here). With MPU you get slighly better errors with slightly more MACS. What happens with ReLU if you increase the MACS a bit, e.g. by doing a few more epochs? Sure, you test ResNet 34 and 50 in which MACS increase. But there is more than one way to increase the MACS...
>
> **A6**: We agree that there is more than one way to increase the MACS of neural networks. However, we observe that in ResNet18, the training process already converges so that the performance gain would be minimal or even negative with more epochs. To support our opinion, we have performed an additional experiment on matching the MACS of different activations on ResNet18 with a larger number of training epochs. The results are reported in Table 5 of the revised manuscript. Conversely, the proposed activation function increases the MACS by only 0.5% (71.17G to 71.56G) but is **theoretically guaranteed to have a stronger expressive capability and better empirical results in most cases.**
>
> Moreover, compared to changing the model to ResNet34 or 50, which almost doubles or even triples the MACS, the modification of the activation function seems a more economical choice.
>
> Thanks again for your questions and comments. Please let us know if you have any more questions!
>
> # References
> [1] He, K., Zhang, X., Ren, S., & Sun, J. Delving Deep into Rectifiers: Surpassing Human-Level Performance on ImageNet Classification.
>
> [2] Shang, W., Sohn, K., Almeida, D., & Lee, H. Understanding and Improving Convolutional Neural Networks via Concatenated Rectified Linear Units.
>
> [3] Dubey, S. R., Singh, S. K., & Chaudhuri, B. B. Activation Functions in Deep Learning: A Comprehensive Survey and Benchmark.
>
> [4] Xiao, C., Zhong, P., & Zheng, C. Enhancing adversarial defense by k-winners-take-all.
>
> [5] Goodfellow, I. J., Warde-Farley, D., Mirza, M., Courville, A., & Bengio, Y. Maxout Networks.

---

> > ### Comment · Reviewer_L1wf · 2023-11-21
> > **Thank you for answers. Review was updated.**
> >
> > Thank you for your detailed answers. I missed the point in Theorem 2, and understood better the motivation for improving the activation function. However, you answer regarding m being small added a new weakness. I updated the review, indicating removed and added "Weaknesses" and more comment on the questions. The overall score was also upgraded.
> >
> > Regarding m: The use of very small m suggested that very little is to gain from MIMO activations. See updated review.
> >
> > Question above using same hyperparameters : You say that "The same hyperparameters were maintained across all experiments, with the sole variation being the Activation Function (AF)." Why? How the results will look if all algorithms use their best hyperparameters?
> >
> > Increasing MACS: OK, doing more epochs might make sense less, but there are other ways. e.g. slightly wider network?
> >
> > Implementing MPU: Still unclear on how exactly the projection itself is implemented.

---

> > > ### Author Response · Authors · 2023-11-22
> > > **Official Response to Reviewer L1wf (Part 1 / 2)**
> > >
> > > Thank you so much for your appreciation of our work and constructive feedback! This means a lot to us. Below are the responses to new questions.
> > >
> > > > Q1: Limited potential improvements: the authors observed that only using very small input-output size (namely, m=2, which replaces one input one output with two input two output) is the best you can do. Going to m=3 and above does not help. This suggest that there is very little to gain from MIMO non-linearity.
> > >
> > > **A1**: Thank you for the valuable question! We hope to further explain the superiority of MIMO functions over SISO functions in the following two aspects:
> > > 1. **The core difference in the expressive capability of neural networks lies in the shift from SISO AFs to MIMO AFs, rather than that between MIMO AFs with different dimensions.** Intuitively, we can compare the expressive capability of the "true" MIMO AFs (those not reducible to element-wise operations), e.g., the projection to the 2-dimensional second-order cone from $\mathbb{R}^2\to\mathbb{R}^2$, with that of the "fake" MIMO AFs, e.g., the 2-dimensional ReLU function. According to the illustration of our paper, we can also view the 2-dimensional ReLU as the projection from $\mathbb{R}^2$to the polygon $\{(x, y)\mid x\geq 0, y\geq 0\}
> > > $. However, this type of element-wise projection fails to capture the intrinsic variable correlations present in "true" MIMO AFs (This can also be observed by the calculation equation of the MPU in Theorem 4). Generally, the superiority of the MIMO AFs over SISO AFs in the sense of expressive capability is theoretically proved in Theorem 2. Moreover, empirical results in Section 4.1 further illustrate the advantages of the "true" MIMO functions.
> > > 2. Indeed, the lower-dimensional MPUs, like the 2D version, tend to outperform their higher-dimensional peers. However, **this is only the case for our MPU, and doesn’t necessarily mean all MIMO AFs will show this pattern.** We still believe that there are a lot more "true" MIMO functions worth exploring, such as the proximal operators in Section 3. We are also not sure whether this phenomenon is also present in other MIMO activation functions. We will continue exploring this direction and try to find the answer behind it!
> > >
> > > > Q2: Still unclear on how exactly the projection itself is implemented.
> > >
> > > **A2**: Sorry for the confusion caused. Indeed, in the previous version of the paper, the implementation of the cones may be less straightforward for the readers to understand, since the corresponding details are mainly introduced in Appendix E and G and we deeply thank you for pointing this out.
> > >
> > > In the revised manuscript, the implementation of the projection is described in the following three aspects, introduced in Section 2.2: Method:
> > > 1. Integration of the MPU into neural networks: We've provided a thorough walkthrough in Section 2.2.
> > > 2. Formulation of m-dimensional MPUs: With the analytical equations at hand, we've made sure that readers can easily translate the MPU into executable code. This part is detailed in Appendix B and E, and we've included references to it in Section 2.2 to streamline the paper's flow.
> > > 3. Parameter selection in MPUs: We address this in Appendix G and have included references in Section 2.2 as well, so readers can navigate the content with ease.

---

> ### Author Response · Authors · 2023-11-22
> **Official Response to Revierwer L1wf (Part 2 / 2)**
>
> > Q3: OK, doing more epochs might make sense less, but there are other ways. e.g. slightly wider network?
>
> **A3**: We agree that with more computational resources available, increasing the width or the depth of the network may be a good way to improve the accuracy of the networks. However, to increase the width of the ResNet, we have to modify the inner structure of each basic block. Since the structure of ResNet is already mature and classic, modifying the inner structure of a single block may destroy its structural relations inside. Instead, we choose to slightly increase the depth of the ResNet18 architecture by adding one more basic block. We believe that this modification can leverage more computational resources and thus increase the used MACS, but preserves the basic structure of the ResNet module.
>
> We've conducted the corresponding experiment, whose results are already updated in Table 6 of Appendix Section F in the revised manuscript. The result shows that increasing the depth of ResNet indeed improves the accuracy of some AFs such as ReLU, but with a larger MACS cost. In contrast, our proposed MPU still achieves the best performance compared to ResNet18+ with other AFs, with a lower MACS cost.
>
> > Q4: The authors use the same hyperparameters across all experiments, with the sole variation being the Activation Function (AF)." Why? And, more importantly, how the results will look if all algorithms use their best hyperparameters?
>
> **A4**: Thank you for your insightful question! The reason why we originally adopted the same hyperparameters across all experiments is for the fairness of the experiments. We controlled all other factors as the same, and let the AFs be the only variation, to clearly observe the impact of the AFs. Similar methods are also adopted in many other benchmark papers, such as Dubey et al. [1].
>
> Indeed, it is necessary to dig into the potential of each AF and adopt their best parameters respectively. However, searching for the best hyperparameters for each AF is extremely time-consuming, and we may not be able to provide the final result before the discussion stage ends.
>
> Thanks again for your time!
>
> # References
>
> [1] Dubey, S. R., Singh, S. K., & Chaudhuri, B. B. Activation Functions in Deep Learning: A Comprehensive Survey and Benchmark.

---

### Author Response · Authors · 2023-11-17
**Official Response to All Reviewers**

We deeply thank the reviewers for their thorough review and valuable feedback. The comments greatly help to improve our work. We individually address the reviewers' comments in the following posts.

We briefly list the manuscript revisions as follows. In the manuscript, the updates are highlighted in blue.

**For writing**, we cleared the misunderstanding and improved the descriptions from three aspects:

1. Add another Section 2.2 to formally introduce our method after the motivation Section 2.1 in the revised manuscript, where we explicitly define the MPU unit in Definition 1 and the $m$-dimensional cone $C_\alpha^{(m)}$ in Definition 2.

2. Discuss the selection of cones in the appendix Section G, where we mainly show the selection of three factors of the cone, including the cone's vertex and axis, the dimension of the cone, and the half-apex angle $\alpha$ of the cone.

3. Add the analysis for computational complexity in the appendix Section H.

Moreover, we have performed **a series of experiments** and added them to the manuscript. The experiments include:

1. More comparison experiments to Section 4, including Winner-takes-all, Maxout, and CReLU.

2. Further investigation on the choice of the cone and its dimensionality.

3. Match the MACS of different activations on ResNet18 with varying training epochs.

4. Comparison experiments on deeper ResNet 101.

We hope our revised manuscript and additional experiments help to address the reviewers' problems. Please let us know if any more improvements can be made.

---

### Author Response · Authors · 2023-11-21

Dear Reviewers and Area Chair,

The end of the discussion phase is approaching.

We deeply thank the reviewers for their insightful comments. We also tried our best to address the reviewers' concerns. However, we are still waiting for replies from reviewers L1wf and WKdp. Could you please take a look at our responses and let us know if any more improvements can be made?

Thanks, Authors

---

### Meta-Review · Area_Chair_uyWK · 2023-12-05

**Metareview:**

A paper proposing use of an activation function, referred to as multivariate projection unit (MPU), with multiple inputs and multiple outputs (MIMO) implementing a projection onto a convex cone, or more generally, a proximal operator (subsuming several existing activations). The proposed activation function is shown to possesses richer expressive power.

All seem to have found the use of a MIMO activation, inspired by concepts used in convex optimization, novel and intriguing. However, assessments remained largely lukewarm despite the discussions with the authors: of the issues common in the reviews some, e.g.: MPU implementation details, organization and writing,... have been addressed to varying extent, but otherwise a major issue concerning more solid evidence for the benefits of using MPU remained. The authors are encouraged to incorporate the important feedback given by the knowledgeable reviewers.

**Justification For Why Not Higher Score:**

Reviewers and I felt that a stronger conviction is needed to turn the idea from nice to useful in practice.

**Justification For Why Not Lower Score:**

N\A

---

### Decision · Program_Chairs · 2024-01-16

Reject